# Unified Surgical World Model for Structured Understanding, Long-Horizon Prediction, and Fine-Grained Generation

## Abstract

World models are capable of learning environment dynamics and supporting long-horizon prediction and data-efficient policy learning by synthesizing plausible rollouts. These models provide a flexible and powerful framework for training agents in environments where data is scarce, annotation is costly, and exploration is constrained. In the context of surgery, the surgical intelligence field faces significant challenges due to the lack of high-quality and diverse multimodal data for training surgical vision-language models, as well as the absence of highly realistic simulators for training surgical robots. Surgical world models address these challenges by both generating multimodal data and serving as a surgical embodied simulator, making them ideal for advancing surgical robotics and intelligence. We propose a Unified Surgical World Model (UniSWM), which unifies structured understanding, long-horizon prediction, and fine-grained generation through a mixture of transformers. UniSWM acts as both a data generator and a simulator for surgical robotics, supporting vision–language and vision–language–action training across in-body and operating room settings. This model integrates structured understanding with discrete action tokens for phase, step, action, and movement, and supports long-horizon prediction for multi-step surgical trajectories. It conditions fine-grained generation on action and movement tokens, aligning frames to deterministic textual descriptions, and eliminates the need for optical flow or kinematic labels. To enable the training of world models, we introduce UniSWM-DB, a diverse multimodal dataset containing 1.81 million samples specifically designed for surgical training. To evaluate the capabilities of UniSWM, we propose UniSWM-Bench, a comprehensive benchmark covering five understanding tasks, two prediction tasks, and three generation tasks. Experimental results demonstrate that UniSWM significantly outperforms existing models, including GPT-5, Gemini-2.5-Pro, and Qwen-VL-Max, excelling in structured understanding, long-horizon prediction, and coherent, controllable visual generation.

## 1 Introduction

World models provide a powerful framework for learning environment dynamics (Li et al., 2025; Zhao et al., 2025a), supporting long-horizon prediction, and enabling data-efficient policy learning by synthesizing plausible rollouts (Ha & Schmidhuber, 2018; Hafner et al., 2019). These capabilities are especially valuable in domains where data is scarce, annotation is costly, and exploration is restricted (Xiang et al., 2023; Ding et al., 2024b). In the context of surgery, the field of surgical intelligence faces significant challenges due to the lack of high-quality, diverse multimodal data necessary for training surgical vision-language models (VLMs), as well as the absence of highly realistic simulators for training surgical robots (Zeng et al., 2025b; Min et al., 2025). Surgical world models address these challenges by both generating multimodal data and serving as an embodied simulator for surgical robotics, enabling realistic and data-efficient training for both VLMs and vision-language-action (VLA) models (Zhao et al., 2025b; Lu et al., 2025).

Despite advances in surgical scene analysis and generative modeling (Khan et al., 2025), current systems remain fragmented, often focusing on isolated tasks such as phase recognition, action prediction, or single-view video generation (Hamoud et al., 2025; Schmidt et al., 2021). These approaches

typically concentrate on in-body videos and fail to provide mechanisms for predicting future states or controlling instrument behavior (Twinanda et al., 2017; Jin et al., 2021; Ho et al., 2020; Rombach et al., 2022; Cho et al., 2024; Iliash et al., 2024; CAMMA Lab). As a result, they lack the capability to comprehensively model surgical workflows and to generate realistic, controllable simulations for surgical robots.

To address these limitations, we propose a Unified Surgical World Model (UniSWM), a framework that unifies structured understanding, long-horizon prediction, and fine-grained generation through a mixture of transformers. It serves dual roles as both a data generator and a simulator for surgical robotics, supporting vision–language and vision–language–action training across in-body and operating room (OR) settings (Paik et al., 2014). UniSWM organizes control using discrete *action tokens*, which capture structured aspects of the surgical workflow. Furthermore, it enables long-horizon prediction by forecasting multi-step Phase and Step trajectories, and supports fine-grained generation conditioned on action and movement tokens. This capability allows UniSWM to produce realistic and coherent visual outputs that evolve according to the surgical context.

To enable the training of world models, we introduce UniSWM-DB, a diverse multimodal dataset containing 1.81 million samples specifically designed for surgical training. We also propose UniSWM-Bench, a comprehensive benchmark covering five understanding tasks, two prediction tasks, and three generation tasks, which allows us to evaluate the model's performance in recognition, controllable synthesis, and long-horizon forecasting within a unified framework.

Our contributions are fivefold:

1. We present UniSWM, a unified surgical world model that integrates structured understanding, long-horizon prediction, and fine-grained generation across in-body and OR scenes.

2. We implement structured understanding through discrete action tokens, which capture the hierarchical aspects of the surgical workflow, enabling effective structured scene analysis.

3. We enable long-horizon prediction by forecasting multi-step Phase and Step trajectories, planning in latent space, and advancing the latent state under predicted actions and movements, decoding boundary frames only when necessary to preserve global coherence.

4. We achieve fine-grained, token-driven generation by conditioning on action and movement tokens, aligning generated frames with deterministic textual descriptions.

5. We propose UniSWM-DB, a diverse multimodal dataset containing 1.81 million samples for training surgical world models, and UniSWM-Bench, a comprehensive evaluation suite covering five understanding tasks, two prediction tasks, and three generation tasks.

## 2 RELATED WORK

**World model** World model learn latent dynamics for imagination-based control, long-horizon forecasting, and data efficiency (Ha & Schmidhuber, 2018; Hafner et al., 2019). Subsequent advances combine stochastic latent rollouts with value learning to stabilize training and improve sample efficiency (Hafner et al., 2019). In parallel, diffusion and latent diffusion substantially improved fidelity and temporal coherence in forward prediction and synthesis (Ho et al., 2020; Rombach et al., 2022; Cao et al., 2024). Vision–language–action (VLA) systems further connect internet-scale knowledge with embodied policies (Ma et al., 2024; Brohan et al., 2023; Driess et al., 2023; Kim et al., 2025). UniSWM situates itself at this intersection by unifying understanding, controllable generation, and long-horizon prediction under a single, domain-tailored framework.

**Surgical Intelligence** Surgical perception has progressed across phase recognition, tool analysis, and workflow modeling (Demir et al., 2023; Ding et al., 2024a), with canonical efforts in endoscopic phase recognition and video modeling (Twinanda et al., 2017; Jin et al., 2021; CAMMA Lab). Community benchmarks expanded evaluation to OR context and instrument segmentation (Escamirosa et al., 2015; Allan et al., 2019; Srivastav et al., 2018), yet joint modeling of in-body and OR signals with forward prediction remains limited. Diffusion-based generation is emerging for surgical content creation and interactive synthesis (Cho et al., 2024; Iliash et al., 2024; Zeng et al., 2025a), but action/movement–conditioned futures that preserve instrument kinematics and tissue realism are underexplored.

Figure 1: Overview of UniSWM, showing the end-to-end pipeline from multimodal perception to structured understanding, long-horizon prediction, and fine-grained generation.

# 3 METHOD

## 3.1 OVERVIEW

UniSWM encodes multimodal observations into a compact latent state, advances this state under a set of action tokens that reflect the surgical hierarchy, and decodes both future states and visuals using a decoder-only foundation model conditioned on specific tokens from the text prompt. The model performs structured understanding, long-horizon prediction, and fine-grained generation. As depicted in Figure 2, the **Left** part demonstrates latent dynamics involving visual, text, and action tokens across Phase, Step, Atomic Action, and Movement. Meanwhile, the **Middle** part presents the foundation of MoT (Mixture of Transformers), which selects the optimal corresponding streams for each task. Moreover, the **Right** part showcases three distinct decoders for the purposes of understanding, prediction, and generation. In the following, we illustrate each module of our model in detail.

**Initial Scene Generation** Before the pipeline begins, the model initializes the scene from a textual prompt: we apply $\mathrm{Tok}(u_0) \xrightarrow{\mathrm{MoT}} \hat{u}_0 \xrightarrow{\mathrm{Dec_{gen}}}$ Initial Scene. This optional step provides a coherent visual state for subsequent reasoning and synthesis.

**Perception Encoders** Perception supplies semantically aligned *vision features* for understanding and high-fidelity *latents* for generation. We denote visual observations by $x_t$ and text instructions by $u_t$. Two complementary encoders are used: a CLIP-style semantic encoder and a VAE-style generative encoder. The CLIP-style semantic encoder maps images and project the image embedding into a sequence of visual tokens consumed by the LLM: $e_t^{\mathrm{img}} = \mathrm{Enc}_{\mathrm{img}}(x_t)$, $s_t^{\mathrm{vis}} = \mathrm{Tok}_{\mathrm{vis}}(e_t^{\mathrm{img}})$. The VAE-style generative encoder compresses frames into latents suitable for high-fidelity synthesis: $v_t = \mathrm{Enc}_{\mathrm{gen}}(x_t)$, $\hat{x}_t = \mathrm{Dec}_{\mathrm{vae}}(v_t)$.

**Latent Dynamics and Action Tokens** A recurrent latent aggregates recent evidence and provides a substrate for imagination: $z_t \sim p_\phi(z_t \mid z_{t-1}, s_t^{\mathrm{vis}}, v_t, a_{t-1})$. Surgical control is expressed with four discrete token sequences mirroring clinical hierarchy: Phase and Step capture workflow progression; Atomic Action describes intended instrument usage; Movement captures short-horizon motion primitives via a vector-quantized codebook over motion patterns: $\tilde{c}_t = \{\tilde{c}_t^{\mathrm{phase}}, \tilde{c}_t^{\mathrm{step}}, \tilde{c}_t^{\mathrm{action}}, \tilde{c}_t^{\mathrm{mv}}\}$.

**Foundation with Mixture of Transformers** We instantiate the foundation model for long-horizon prediction and fine-grained generation using a Mixture-of-Transformers (MoT) architecture. First, a shared multi-modal stem computes contextualized hidden states for the entire serialized context: $H_t^{(0)} = \mathrm{Stem}_\theta(\mathcal{C}_t, z_t)$, where $\mathrm{Stem}_\theta$ is a stack of self-attention and MLP blocks with unified positional encoding for all token types. After this, two *full Transformer branches* specialize the representation for language understanding/planning and visual planning respectively: $H_t^{\mathrm{und}} = \mathrm{Transf}_{\psi_{\mathrm{und}}}(H_t^{(0)})$, $\qquad H_t^{\mathrm{gen}} = \mathrm{Transf}_{\psi_{\mathrm{gen}}}(H_t^{(0)})$. A lightweight router then produces

mixture weights for text and image streams (or selects a top-1 branch). With a pooled summary $\text{Pool}(H_t^{(0)})$, we compute

$$\alpha^{\text{text}} = \text{Softmax}\big(W_{\text{r}}^{\text{text}} \cdot \text{Pool}(H_t^{(0)})\big), \quad \alpha^{\text{img}} = \text{Softmax}\big(W_{\text{r}}^{\text{img}} \cdot \text{Pool}(H_t^{(0)})\big) \in \mathbb{R}^2.$$

The branch-specific hidden states for each stream are

$$\bar{H}_t^{\text{text}} = \alpha_1^{\text{text}} H_t^{\text{und}} + \alpha_2^{\text{text}} H_t^{\text{gen}}, \qquad \bar{H}_t^{\text{img}} = \alpha_1^{\text{img}} H_t^{\text{und}} + \alpha_2^{\text{img}} H_t^{\text{gen}}.$$

Finally, separate vocabularies/Codebooks decode the two streams in *one forward pass*:

$$F_\theta^{\text{text}}(\mathcal{C}_t, z_t) = \text{AR}\big(\text{Softmax}(W_{\text{text}} \bar{H}_t^{\text{text}})\big), \qquad F_\theta^{\text{image}}(\mathcal{C}_t, z_t) = \text{AR}\big(\text{Softmax}(W_{\text{img}} \bar{H}_t^{\text{img}})\big).$$

"interleaved" refer to serializing the two streams into one sequence with sentinel tags, $\langle\text{TEXT}\rangle \hat{s} \langle/\text{TEXT}\rangle \langle\text{IMG}\rangle \hat{u} \langle/\text{IMG}\rangle$, while routing ensures each segment predominantly uses its appropriate branch. At test time, decoding proceeds in a single pass with shared KV-cache: (a) generate the text stream (greedy or nucleus sampling) until $\texttt{<EOS\_TEXT>}$; (b) generate the image/clip planning tokens until $\texttt{<EOS\_IMG>}$. Both streams reuse the stem states and differ only in the routed branch and output vocabulary. There is no speculative multi-pass decoding or latent rollout here, and the produced text tokens or visual tokens are then consumed by the understanding decoder, prediction decoder and generation decoder.

**Understanding Decoder for Structured Scene Analysis**  We reinterpret near-term text tokens to estimate the current surgical state. At time $t$, the understanding decoder produces *six* textual heads: phase, step, action, triplet, grounding, and environment. Each head is a short token sequence generated under teacher forcing:

$$\big(\hat{s}_t^{\text{phase}}, \hat{s}_t^{\text{step}}, \hat{s}_t^{\text{action}}, \hat{s}_t^{\text{trip}}, \hat{s}_t^{\text{grd}}, \hat{s}_t^{\text{env}}\big) = \text{Dec}_{\text{und}}(\hat{s}_t). \tag{1}$$

Here, *Phase* is the workflow stage, *Step* refines the phase, *Action* denotes the atomic operation, and *Triplet* consists of "(instrument,verb ,target)". On the other hand, *Grounding* outputs $[x_1, y_1, x_2, y_2] \in [0,1]^4$ (optionally quantized), and *Environment* briefly describes residual factors (e.g., visibility, bleeding). The target strings are $s_t^{\text{phase}}, s_t^{\text{step}}, s_t^{\text{action}}, s_t^{\text{trip}}, s_t^{\text{grd}}, s_t^{\text{env}}$. With tokenizer $\text{Tok}(\cdot)$ and vocabulary $\mathcal{V}$, we train the understanding decoder with token-level cross-entropy:

$$\mathcal{L}_{\text{und}} = \sum_{h \in \{\text{phase},\text{step},\text{action},\text{trip},\text{grd},\text{env}\}} \sum_{\ell=1}^{L_h} \text{CE}\Big(w_\ell^{(h)}, \hat{p}_\ell^{(h)}\big(\cdot \mid w_{<\ell}^{(h)}\big)\Big). \tag{2}$$

**Prediction Decoder for Long-horizon States**  Given the current latent $z_t$ and the textual summaries, the prediction decoder generates a horizon of workflow states as text. Formally, it's conditioned on $z_t$, $\hat{s}_t^{\text{phase}}$, and $\hat{s}_t^{\text{step}}$, then outputs two serialized sequences for $[t+1, t+H]$:

$$\big(\hat{s}_{t+1:t+H}^{\text{phase}}, \hat{s}_{t+1:t+H}^{\text{step}}\big) = \text{Dec}_{\text{pred}}\big(z_t, \hat{s}_t^{\text{phase}}, \hat{s}_t^{\text{step}}\big). \tag{3}$$

Denote the ground-truth strings by $s_{t+1:t+H}^{\text{phase}}$ and $s_{t+1:t+H}^{\text{step}}$, training of the prediction decoder uses token-level cross-entropy over the serialized horizons.

**Generation Decoder for Fine-grained Visuals**  The generation decoder synthesizes a *single* target frame at a specified look-ahead $k$ using only information available at time $t$. The design choice of the generation decoder can be quite flexible, and in this article, we apply the latent diffusion (flow) based method, which consists of a conditional generation model and an auto-encoder. More precisely, the generation process is controlled by $z_t$, on image-side planning tokens $\hat{u}_{t+\Delta k}$ predicted at time $t$, and on planned control tokens $\tilde{c}_{t+\Delta k}^{\text{action}}$ and $\tilde{c}_{t+\Delta k}^{\text{mv}}$:

$$\hat{v}_{t+\Delta k} \sim \text{Dec}_{\text{gen}}\big(z_t, \hat{u}_{t+\Delta k}, \tilde{c}_{t+\Delta k}^{\text{action}}, \tilde{c}_{t+\Delta k}^{\text{mv}}\big), \qquad \hat{x}_{t+\Delta k} = \text{Dec}_{\text{vae}}\big(\hat{v}_{t+\Delta k}\big), \tag{4}$$

for $\Delta k \in \{1, \dots, H\}$. Note that there is no autoregressive rollout over future frames: each $\hat{x}_{t+\Delta k}$ is generated directly from the context at time $t$. For training, the visual generation loss (denoted by $\mathcal{L}_{\text{gen}}$), corresponds to the standard flow matching loss.

**Training Objective** The final objective aggregates all the components defined above: understanding, prediction, and generation:

$$\mathcal{L} = \gamma \, \mathcal{L}_{\text{und}} + \alpha \, \mathcal{L}_{\text{pred}} + \beta \, \mathcal{L}_{\text{gen}},$$

where $\mathcal{L}_{\text{und}}$, $\mathcal{L}_{\text{pred}}$, and $\mathcal{L}_{\text{gen}}$ are the losses for understanding, prediction, and generation, respectively.

## 4 EXPERIMENTS

**Dataset** We evaluate the performance of the UniSWM model using the UniSWM-DB dataset, a comprehensive multimodal dataset designed to support the training and evaluation of models in the context of surgical intelligence. As shown in Figure 1, the dataset consists of a variety of tasks across three main categories: Generation, Prediction, and Understanding. Specifically, it includes tasks such as Caption Control (Out-of-body), Movement & Action Control (In-body), Phase Prediction (In-body), Step Prediction (In-body), as well as various Understanding tasks like Action Recognition, Caption Generation, Safety Recognition, and Visual Question Answering, among others. The dataset contains a total of over 1.81 million samples, with each task and sub-task having its own distinct distribution and data partitioning.

**Benchmark** UniSWM-Bench serves as the evaluation benchmark for UniSWM and includes a task distribution designed to test the model's ability to handle a wide range of surgical tasks. The benchmark is divided into three major categories: Understanding (5 Tasks), Prediction (2 Tasks), and Generation (3 Tasks). As shown in Table 2, tasks within the Understanding category include Phase Recognition, Action Recognition, Triplet Recognition, Instrument Grounding, and Environment Recognition, with sample sizes ranging from 0.5K to 1K per task. The Prediction category includes Phase Prediction and Step Prediction, with 4K samples per task. For the Generation category, the benchmark evaluates Scene Initialization and Scene Evolution, both with 0.5K samples each. These tasks span both in-body and out-of-body surgical contexts and are critical for assessing the model's performance in terms of both real-time understanding and long-horizon predictions.

Table 1: UniSWM-DB task distribution.

| Type | Task & Generation Condition | Domain | Count |
|---|---|---|---|
| Generation | Caption Control | Out-of-body | 109K |
| Generation | Movement & Action Control | In-body | 171K |
| Prediction | Phase Prediction | In-body | 40K |
| Prediction | Step Prediction | In-body | 40K |
| Understanding | Action Recognition | In-body | 390K |
| Understanding | Caption Generation | Out-of-body | 164K |
| Understanding | Caption Generation | In-body | 306K |
| Understanding | Phase Recognition | In-body | 311K |
| Understanding | Safety Recognition | In-body | 110K |
| Understanding | Triplet Recognition | In-body | 103K |
| Understanding | Visual Question Answering | In-body | 11K |

Table 2: UniSWM-Bench task distribution.

| Type | Domain | Task | Count |
|---|---|---|---|
| Understanding | In-body | Phase Recognition | 1K |
| | In-body | Action Recognition | 1K |
| | In-body | Triplet Recognition | 1K |
| | In-body | Instrument Grounding | 1K |
| | In-body | Environment Recognition | 0.5K |
| Prediction | In-body | Phase Prediction | 4K |
| | In-body | Step Prediction | 4K |
| Generation | In-body | Scene Initialization | 0.5K |
| | In-body | Scene Evolution | 0.5K |
| | Out-of-body | Scene Initialization | 0.5K |

### 4.1 EVALUATION OF STRUCTURED UNDERSTANDING

In this evaluation, we compare a total of 29 models, including open-source variants, and conduct stability tests (see Appendix for details). However, the main comparison focuses on two strong general-purpose Vision-Language Models (VLMs), Gemini-2.5-Pro and GPT-5, using the UniSWM-Bench protocol across various tasks. These tasks assess the model's ability to understand and process structured understanding of the surgical workflow, including workflow recognition, compositional semantics recognition, spatial perception, and environment answering (see Table 3).

**Workflow Recognition** The first crucial step in structured understanding is workflow recognition, which involves recognizing phases and actions within the surgical procedure. General VLMs show moderate performance on this task, with GPT-5 achieving accuracy, precision, recall, and Jaccard scores of 28.30/0.65/0.70/7.18, and Gemini-2.5-Pro showing 28.30/16.23/13.95/6.06. These models struggle with the finer nuances of phase transitions and action timings due to their reliance on instruction-following mechanisms without deeper temporal reasoning. UniSWM, by contrast, leverages latent dynamics with action-aware conditioning, achieving 81.90/71.79/67.59/54.42 on phase

Table 3: Evaluation of structured understanding on UniSWM-Bench across five tasks.

| Model | Phase Recognition | | | | Action Recognition | | | | Triplet Recognition | | | | Instrument Grounding | | | | Environment Answering | | | |
|---|---|---|---|---|---|---|---|---|---|---|---|---|---|---|---|---|---|---|---|---|
| | Acc. | Pre. | Rec. | Jac. | Acc. | Pre. | Rec. | Jac. | Acc. | Ins. | Ver. | Tar. | mIoU | mAP$_{50}$ | mAP$_{75}$ | AP | Acc. | B4 | MET | R1 |
| SmolVLM2-2.2B | 20.81 | 12.18 | 15.37 | 6.35 | 14.93 | 13.05 | 12.25 | 5.61 | 0.00 | 7.87 | 3.71 | 0.62 | 2.89 | 0.27 | 0.07 | 0.08 | 23.65 | 3.03 | 12.19 | 15.60 |
| Skywork-R1V-38B | 6.94 | 17.24 | 14.31 | 1.56 | 11.83 | 14.42 | 12.62 | 2.76 | 0.12 | 21.06 | 9.33 | 2.25 | 9.86 | 1.60 | 0.00 | 0.28 | 35.63 | 0.32 | 1.36 | 1.56 |
| Phi4-Multimodal | 17.92 | 14.14 | 15.50 | 5.36 | 24.37 | 13.35 | 12.38 | 5.74 | 0.29 | 11.50 | 12.58 | 3.87 | 1.17 | 0.13 | 0.00 | 0.04 | 34.25 | 0.34 | 1.48 | 1.74 |
| Mistral-Small-24B | 25.60 | 26.80 | 25.92 | 11.64 | 12.50 | 5.73 | 12.68 | 1.78 | 0.50 | 9.08 | 8.79 | 5.96 | 17.64 | 5.93 | 0.27 | 1.45 | 36.79 | 0.15 | 0.59 | 0.71 |
| PaliGemma2-3B | 7.29 | 10.08 | 13.90 | 2.35 | 11.57 | 13.92 | 12.58 | 2.26 | 0.00 | 0.96 | 2.19 | 3.06 | 0.01 | 0.00 | 0.00 | 0.00 | 31.78 | 1.55 | 5.65 | 9.04 |
| Llama-4-Scout-17B-16E | 40.70 | 27.96 | 18.02 | 8.71 | 30.97 | 19.46 | 12.75 | 4.66 | 0.21 | 3.35 | 3.56 | 7.46 | 36.72 | 35.73 | 7.17 | 12.99 | 36.99 | 0.04 | 0.20 | 0.33 |
| Kimi-VL-A3B-Instruct | 35.00 | 20.79 | 19.04 | 10.25 | 25.23 | 12.10 | 14.17 | 6.91 | 0.02 | 12.04 | 9.83 | 2.14 | 9.63 | 4.93 | 0.73 | 1.62 | 32.11 | 1.65 | 6.62 | 8.60 |
| Kimi-VL-A3B-Thinking | 6.53 | 16.61 | 14.31 | 1.06 | 11.23 | 14.07 | 12.90 | 2.33 | 0.04 | 6.50 | 9.21 | 1.08 | 9.36 | 1.90 | 0.13 | 0.48 | 34.09 | 0.34 | 1.44 | 1.74 |
| Gemma3-27B | 14.03 | 25.37 | 16.93 | 4.52 | 31.97 | 20.60 | 13.50 | 5.01 | 0.13 | 10.00 | 6.35 | 1.67 | 17.45 | 6.17 | 0.10 | 1.41 | 36.02 | 0.03 | 0.13 | 0.26 |
| MiMo-VL-7B-SFT | 20.90 | 20.73 | 16.46 | 7.15 | 24.37 | 17.56 | 15.72 | 7.40 | 0.13 | 7.98 | 5.10 | 4.85 | 33.45 | 28.57 | 2.80 | 8.55 | 35.40 | 0.09 | 0.37 | 0.66 |
| MiMo-VL-7B-RL | 25.73 | 15.73 | 14.77 | 6.77 | 26.90 | 14.21 | 14.04 | 7.05 | 0.21 | 6.25 | 5.14 | 2.14 | 36.86 | 34.77 | 3.77 | 10.78 | 34.97 | 0.10 | 0.40 | 0.74 |
| MiniCPM-V-2.6 | 16.85 | 14.24 | 14.20 | 6.10 | 24.43 | 14.14 | 13.42 | 7.41 | 0.04 | 17.02 | 13.93 | 1.35 | 17.87 | 7.33 | 0.57 | 1.87 | 33.32 | 1.14 | 5.50 | 6.18 |
| MiniCPM-o-2.6 | 15.10 | 23.77 | 20.63 | 8.68 | 29.67 | 12.60 | 12.47 | 5.22 | 0.35 | 23.31 | 8.92 | 5.29 | 20.33 | 9.80 | 1.10 | 2.96 | 35.29 | 0.24 | 0.74 | 1.15 |
| Qwen-Omni-3B | 34.16 | 18.11 | 18.27 | 10.59 | 28.23 | 12.76 | 12.38 | 5.69 | 0.00 | 10.33 | 7.06 | 1.31 | 22.17 | 16.20 | 3.47 | 6.25 | 35.03 | 0.14 | 0.87 | 0.73 |
| Qwen-Omni-7B | 22.01 | 18.16 | 18.10 | 7.66 | 29.93 | 29.58 | 13.20 | 5.81 | 0.15 | 10.79 | 6.87 | 5.46 | 34.60 | 40.53 | 5.63 | 14.17 | 37.45 | 0.26 | 1.06 | 1.24 |
| Qwen2.5-VL-7B | 27.57 | 20.50 | 18.48 | 8.33 | 31.17 | 5.82 | 12.55 | 4.02 | 0.23 | 9.92 | 4.65 | 5.21 | 10.63 | 0.97 | 0.03 | 0.26 | 37.17 | 4.43 | 15.14 | 13.27 |
| Qwen2.5-VL-32B | 44.77 | 26.83 | 22.05 | 13.63 | 31.67 | 31.84 | 13.72 | 5.34 | 0.23 | 26.22 | 6.69 | 2.10 | 11.36 | 1.93 | 1.07 | 1.03 | 42.43 | 0.21 | 2.08 | 1.74 |
| Qwen2.5-VL-72B | 37.03 | 23.75 | 20.21 | 10.82 | 28.83 | 16.52 | 13.42 | 6.21 | 0.19 | 26.33 | 8.16 | 4.58 | 42.70 | 45.95 | 25.75 | 25.69 | 41.89 | 0.24 | 1.62 | 2.37 |
| InternVL3-8B | 30.93 | 22.21 | 19.38 | 10.71 | 29.30 | 14.16 | 13.03 | 6.36 | 1.65 | 51.49 | 8.87 | 24.39 | 22.49 | 7.13 | 0.20 | 1.58 | 34.77 | 0.98 | 4.37 | 4.35 |
| InternVL3-78B | 33.17 | 34.01 | 25.25 | 15.05 | 28.77 | 21.24 | 12.69 | 6.22 | 0.37 | 37.39 | 8.54 | 3.44 | 29.41 | 18.20 | 1.60 | 4.99 | 36.70 | 0.16 | 1.39 | 0.92 |
| MedVLM-R1 | 10.50 | 12.54 | 16.15 | 2.40 | 31.13 | 16.39 | 12.52 | 3.94 | 0.02 | 45.72 | 8.98 | 0.31 | 5.31 | 0.30 | 0.00 | 0.03 | 31.25 | 2.42 | 7.66 | 10.85 |
| Lingshu-7B | 39.77 | 13.63 | 23.53 | 9.07 | 31.50 | 16.41 | 12.94 | 4.33 | 0.08 | 2.15 | 7.98 | 3.65 | 28.72 | 26.13 | 3.37 | 8.20 | 35.41 | 0.71 | 2.74 | 3.02 |
| Lingshu-32B | 36.31 | 23.98 | 22.22 | 12.85 | 26.60 | 20.07 | 17.96 | 10.09 | 0.21 | 21.04 | 7.17 | 3.77 | 26.03 | 18.20 | 2.90 | 5.89 | 34.69 | 0.11 | 0.59 | 0.74 |
| MedGemma-4B | 27.07 | 18.53 | 23.33 | 9.63 | 22.50 | 4.42 | 12.32 | 3.53 | 0.00 | 2.89 | 1.42 | 1.43 | 9.43 | 0.57 | 0.00 | 0.08 | 37.09 | 0.05 | 0.32 | 0.58 |
| MedGemma-27B | 43.40 | 24.68 | 15.30 | 7.37 | 29.90 | 20.73 | 12.38 | 4.73 | 0.13 | 8.39 | 3.65 | 3.62 | 17.20 | 2.23 | 0.00 | 0.44 | 35.98 | 0.05 | 0.24 | 0.44 |
| SurgVLM-72B | 73.05 | 64.91 | 65.30 | 49.10 | 45.13 | 30.05 | 31.88 | 18.60 | 4.91 | 47.66 | 12.91 | 38.91 | 59.34 | 74.20 | 28.30 | 36.28 | 69.52 | 46.65 | 58.16 | 74.58 |
| Qwen-VL-Max-Latest | 43.40 | 6.20 | 14.29 | 6.20 | 24.90 | 9.38 | 11.95 | 6.04 | 2.32 | 75.53 | 9.18 | 71.84 | 8.53 | 0.30 | 0.00 | 0.05 | 34.50 | 0.64 | 0.53 | 0.64 |
| GPT-5-0807-Global | 28.30 | 0.65 | 0.70 | 7.18 | 14.20 | 6.36 | 8.00 | 2.79 | 1.90 | 54.85 | 9.28 | 44.83 | 1.92 | 1.20 | 0.40 | 0.50 | 36.98 | 1.72 | 1.61 | 1.72 |
| Gemini-2.5-Pro-06-17 | 28.30 | 16.23 | 13.95 | 6.06 | 25.60 | 9.34 | 12.21 | 4.94 | 0.00 | 45.06 | 34.24 | 8.20 | 27.92 | 15.90 | 1.60 | 4.90 | 34.37 | 2.44 | 2.08 | 2.44 |
| **UniSWM** | **81.90** | **71.79** | **67.59** | **54.42** | **63.40** | **57.44** | **42.77** | **31.34** | **47.51** | **71.81** | **64.56** | **49.77** | **88.11** | **94.20** | **88.70** | **73.75** | **69.85** | **92.83** | **97.24** | **94.17** |

recognition and 63.40/57.44/42.77/31.34 on action recognition. The significant performance improvement emphasizes the importance of structured understanding for temporal modeling in surgical workflows, which general VLMs cannot fully capture.

**Compositional Semantics Recognition** The ability to reason compositely over instruments, verbs, and targets is essential for understanding the context of a surgical procedure. General VLMs such as GPT-5 achieve only 1.90% triplet accuracy, with factor-wise scores of 54.85/9.28/44.83 for instrument/verb/target, while Gemini-2.5-Pro yields 0.00% accuracy and factor-wise scores of 45.06/34.24/8.20. This indicates that these models struggle with maintaining global consistency across the components of surgical actions. UniSWM significantly improves on this by achieving 47.51% triplet accuracy and enhancing factor-wise scores to 71.81/64.56/49.77. The discrepancy between the factor-wise and joint triplet scores in general VLMs highlights their weak semantic coherence. UniSWM's structured approach enforces compatibility among participants (instruments), predicates (verbs), and entities (targets), providing a more integrated and semantically consistent understanding of the surgical action.

**Spatial Perception** Accurate spatial perception is essential in surgery, particularly for tasks such as instrument localization and environment interaction. For instrument grounding, GPT-5 achieves only 1.92 mIoU, 1.20 mAP$_{50}$, 0.40 mAP$_{75}$, and 0.50 AP. Gemini-2.5-Pro fares slightly better with 27.92 mIoU, 15.90 mAP$_{50}$, 1.60 mAP$_{75}$, and 4.90 AP. However, UniSWM outperforms both with substantial margins, reaching 88.11 mIoU, 94.20 mAP$_{50}$, 88.70 mAP$_{75}$, and 73.75 AP. These results reveal that traditional VLMs, which are designed for generic visual question answering, fail to handle the dense spatial localization required in surgical tasks. UniSWM's approach ties temporal states with spatial entities, resulting in more consistent and accurate pixel-level predictions that align with the complex spatial relationships in the operating room.

**Environment Answering** Finally, environment answering assesses the model's ability to generate coherent, linguistically rich responses based on the surgical scene. General VLMs show moderate recognition accuracy but poor language quality. GPT-5 achieves 36.98% accuracy, with BLEU-4 of 1.72, METEOR of 1.61, and ROUGE-1 of 1.72, while Gemini-2.5-Pro performs slightly better with 34.37% accuracy and BLEU-4 of 2.44. However, both models struggle to generate fluent and accurate responses due to their focus on either recognition or language quality, but not both. UniSWM, by contrast, achieves 69.85% accuracy, with significantly higher language generation metrics: BLEU-4 of 92.83, METEOR of 97.24, and ROUGE-1 of 94.17. This demonstrates that

UniSWM not only recognizes the environmental context accurately but also generates linguistically coherent responses, effectively closing the loop of structured understanding by ensuring that the perceived scene and the language generated are both faithful to the task and its context.

Table 4: Long-horizon prediction evaluation in UniSWM-Bench. $H$ indicates the horizon between current time and target timestamp. Top: comparison with models trained on UniSWM-DB. Bottom: comparison with mainstream commercial models.

| Model | $H = 1$s | | | | | $H = 5$s | | | | | $H = 30$s | | | | | $H = 60$s | | | | |
|---|---|---|---|---|---|---|---|---|---|---|---|---|---|---|---|---|---|---|---|---|
| | Acc. | Pre. | Rec. | F1 | Jac. | Acc. | Pre. | Rec. | F1 | Jac. | Acc. | Pre. | Rec. | F1 | Jac. | Acc. | Pre. | Rec. | F1 | Jac. |
| **Phase** | | | | | | | | | | | | | | | | | | | | |
| InternVL3-8B | 18.30 | 12.09 | 10.84 | 10.44 | 5.79 | 20.80 | 14.25 | 14.13 | 12.98 | 7.47 | 21.10 | 13.33 | 14.23 | 12.89 | 7.31 | 26.50 | 20.40 | 18.87 | 18.54 | 10.80 |
| Gemma-3-27B | 9.90 | 8.90 | 9.60 | 7.71 | 4.10 | 9.20 | 6.70 | 9.32 | 7.54 | 3.99 | 9.10 | 6.54 | 8.90 | 7.19 | 3.81 | 8.60 | 15.32 | 8.84 | 7.42 | 3.92 |
| Gemma3-4B | 21.00 | 17.87 | 12.34 | 11.14 | 6.30 | 22.60 | 13.42 | 12.20 | 10.66 | 6.04 | 20.40 | 8.51 | 10.49 | 8.57 | 4.83 | 23.40 | 11.67 | 9.21 | 9.05 | 5.13 |
| Qwen2.5-VL-7B | 20.50 | 13.17 | 14.41 | 13.04 | 7.42 | 19.90 | 14.41 | 15.14 | 13.48 | 7.64 | 17.80 | 14.88 | 12.65 | 12.29 | 6.86 | 20.30 | 16.05 | 15.22 | 14.47 | 8.04 |
| LLaVA1.5-7B | 20.00 | 12.64 | 13.18 | 11.55 | 6.51 | 16.80 | 10.63 | 11.01 | 9.49 | 5.23 | 20.20 | 17.29 | 11.40 | 9.93 | 5.53 | 24.20 | 19.50 | 14.46 | 14.39 | 8.16 |
| SurgVLM-7B | 18.10 | 19.24 | 11.90 | 9.50 | 5.35 | 16.00 | 13.21 | 12.85 | 11.77 | 6.51 | 19.80 | 18.42 | 15.09 | 13.77 | 7.69 | 18.10 | 14.79 | 13.97 | 13.66 | 7.55 |
| **UniSWM** | **63.20** | **76.10** | **69.14** | **66.84** | **54.81** | **65.60** | **71.02** | **72.58** | **66.54** | **52.56** | **52.80** | **60.06** | **63.02** | **58.11** | **42.65** | **56.80** | **58.58** | **54.19** | **52.16** | **39.74** |
| **Step** | | | | | | | | | | | | | | | | | | | | |
| InternVL3-8B | 17.50 | 8.83 | 7.06 | 6.84 | 3.74 | 18.60 | 8.20 | 9.17 | 8.08 | 4.47 | 18.40 | 4.57 | 6.18 | 5.04 | 2.84 | 22.30 | 10.93 | 10.65 | 9.41 | 5.29 |
| Gemma-3-27B | 5.70 | 4.82 | 2.67 | 1.92 | 1.01 | 6.90 | 2.13 | 2.53 | 2.00 | 1.07 | 7.20 | 1.92 | 3.26 | 2.08 | 1.10 | 5.50 | 1.82 | 1.93 | 1.42 | 0.75 |
| Gemma3-4B | 19.80 | 13.04 | 8.20 | 7.71 | 4.27 | 20.30 | 7.97 | 6.53 | 6.07 | 3.38 | 20.10 | 5.50 | 6.02 | 5.14 | 2.88 | 22.60 | 9.30 | 7.28 | 7.20 | 4.05 |
| Qwen2.5-VL-7B | 20.30 | 8.27 | 9.23 | 8.35 | 4.71 | 16.40 | 7.90 | 7.94 | 7.20 | 3.96 | 18.20 | 7.62 | 7.99 | 7.09 | 3.93 | 18.40 | 7.64 | 8.25 | 7.23 | 4.00 |
| LLaVA1.5-7B | 17.10 | 7.26 | 6.41 | 6.14 | 3.41 | 16.60 | 4.16 | 5.38 | 4.55 | 2.54 | 20.60 | 8.19 | 6.07 | 5.31 | 3.01 | 23.70 | 7.72 | 8.46 | 7.65 | 4.38 |
| SurgVLM-7B | 18.90 | 8.49 | 4.85 | 4.26 | 2.40 | 15.70 | 7.89 | 6.16 | 5.84 | 3.18 | 17.60 | 8.43 | 5.96 | 5.32 | 2.92 | 18.10 | 5.25 | 6.10 | 5.26 | 2.91 |
| **UniSWM** | **49.60** | **23.64** | **31.50** | **24.25** | **19.02** | **52.00** | **23.00** | **42.97** | **28.35** | **20.92** | **64.80** | **30.51** | **39.30** | **30.08** | **25.25** | **55.20** | **30.09** | **41.56** | **31.37** | **25.39** |
| **Phase&Step** | | | | | | | | | | | | | | | | | | | | |
| InternVL3-8B | 13.60 | 3.61 | 3.42 | 2.91 | 1.58 | 13.80 | 3.48 | 3.93 | 3.25 | 1.78 | 14.40 | 2.11 | 3.69 | 2.34 | 1.30 | 16.70 | 6.01 | 5.26 | 4.73 | 2.61 |
| Gemma-3-27B | 1.80 | 2.66 | 2.75 | 1.23 | 0.65 | 2.00 | 1.06 | 1.13 | 0.93 | 0.48 | 2.00 | 0.80 | 1.54 | 0.86 | 0.44 | 1.60 | 0.88 | 0.52 | 0.50 | 0.26 |
| Gemma3-4B | 15.50 | 6.31 | 3.73 | 3.26 | 1.78 | 17.60 | 4.87 | 2.71 | 2.31 | 1.28 | 17.90 | 3.04 | 3.01 | 2.37 | 1.31 | 18.00 | 3.37 | 3.50 | 3.14 | 1.74 |
| Qwen2.5-VL-7B | 14.00 | 3.61 | 4.29 | 3.58 | 1.97 | 12.00 | 3.32 | 3.69 | 2.90 | 1.57 | 13.50 | 3.92 | 4.02 | 3.51 | 1.92 | 13.20 | 4.48 | 4.71 | 3.83 | 2.07 |
| LLaVA1.5-7B | 13.60 | 3.75 | 3.34 | 3.03 | 1.67 | 12.20 | 2.07 | 2.38 | 1.81 | 0.99 | 17.00 | 4.17 | 2.77 | 2.39 | 1.32 | 18.10 | 4.61 | 3.90 | 3.64 | 2.04 |
| SurgVLM-7B | 12.90 | 3.74 | 2.85 | 1.66 | 0.93 | 10.60 | 4.69 | 2.80 | 2.77 | 1.49 | 12.80 | 4.03 | 3.39 | 2.56 | 1.37 | 10.90 | 2.99 | 2.98 | 2.56 | 1.37 |
| **UniSWM** | **31.00** | **10.19** | **9.44** | **8.49** | **5.97** | **36.60** | **13.42** | **12.31** | **10.67** | **7.57** | **35.40** | **9.97** | **10.59** | **9.34** | **6.62** | **33.30** | **11.67** | **9.78** | **8.43** | **5.79** |
| **Phase** | | | | | | | | | | | | | | | | | | | | |
| Qwen-VL-Max | 44.72 | 33.83 | 32.79 | 31.11 | 24.89 | 51.20 | 34.97 | 31.39 | 30.67 | 24.54 | 55.74 | 40.58 | 44.12 | 39.71 | 31.77 | 41.80 | 31.37 | 35.10 | 31.96 | 25.56 |
| GPT-5 | 52.42 | 29.14 | 38.46 | 32.94 | 26.35 | 51.64 | 25.65 | 34.81 | 29.06 | 23.24 | 55.65 | 26.74 | 37.11 | 30.22 | 24.18 | 39.84 | 21.81 | 35.07 | 26.00 | 20.80 |
| Gemini-2.5-Pro | 39.52 | 23.81 | 38.27 | 25.65 | 20.52 | 43.20 | 28.52 | 42.08 | 31.68 | 25.34 | 49.60 | 25.19 | 31.73 | 26.59 | 21.27 | 45.16 | 27.53 | 31.45 | 25.52 | 20.41 |
| **UniSWM** | **63.20** | **76.10** | **69.14** | **66.84** | **54.81** | **65.60** | **71.02** | **72.58** | **66.54** | **52.56** | **52.80** | **60.06** | **63.02** | **58.11** | **42.65** | **56.80** | **58.58** | **54.19** | **52.16** | **39.74** |
| **Step** | | | | | | | | | | | | | | | | | | | | |
| Qwen-VL-Max | 41.60 | 17.66 | 33.58 | 19.65 | 15.72 | 39.20 | 14.92 | 26.71 | 17.06 | 13.65 | 49.60 | 19.77 | 31.18 | 21.07 | 16.85 | 45.60 | 20.09 | 30.09 | 18.09 | 14.47 |
| GPT-5 | 54.47 | 34.65 | 48.13 | 34.79 | 27.83 | 49.60 | 33.85 | 42.21 | 30.78 | 24.62 | 54.47 | 28.36 | 39.91 | 26.30 | 21.04 | 52.80 | 33.93 | 46.64 | 32.20 | 25.76 |
| Gemini-2.5-Pro | 41.60 | 18.62 | 22.58 | 19.53 | 15.63 | 42.40 | 24.89 | 31.24 | 25.18 | 20.14 | 31.45 | 16.67 | 20.15 | 15.98 | 12.79 | 38.71 | 18.94 | 22.78 | 19.02 | 15.22 |
| **UniSWM** | **49.60** | **23.64** | **31.50** | **24.25** | **19.02** | **52.00** | **23.00** | **42.97** | **28.35** | **20.92** | **64.80** | **30.51** | **39.30** | **30.08** | **25.25** | **55.20** | **30.09** | **41.56** | **31.37** | **25.39** |

## 4.2 Evaluation of Long-Horizon Prediction

**Phase prediction across horizons.** As shown in Table 4, UniSWM leads by large margins at every horizon. At $H$=1s it attains 63.20/76.10/69.14/66.84/54.81 (Acc./Pre./Rec./F1/Jac.), while the best baseline reaches only 21.00 Acc., 13.04 F1, and 7.42 Jac., yielding improvements of +42.20, +53.80, and +47.39 points. Performance remains strong as the look-ahead extends: 65.60/66.54/52.56 at 5s, 52.80/58.11/42.65 at 30s, and 56.80/52.16/39.74 at 60s (Acc./F1/Jac.). Even at 60s, UniSWM exceeds the best baseline by large margins, indicating controlled error accumulation and reliable temporal credit assignment.

**Step prediction and fine-grained stability.** Step labels change more frequently and have higher cardinality, yet UniSWM maintains clear state tracking. Acc. rises from 49.60 at 1s to 52.00 at 5s, peaks at 64.80 at 30s, and remains 55.20 at 60s. F1 and Jac. improve with horizon (24.25 → 28.35 → 30.08 → 31.37 and 19.02 → 20.92 → 25.25 → 25.39), suggesting that medium-range windows expose regularities that UniSWM exploits without sacrificing long-range stability. Recall gains are particularly large relative to baselines, while precision remains competitive.

**Joint Phase&Step prediction.** Joint decoding requires mutual consistency across the workflow hierarchy. UniSWM achieves 31.00/36.60/35.40/33.30 Acc. at $H$=1/5/30/60s and delivers the highest F1 and Jac. at each horizon (e.g., 10.67 F1 and 7.57 Jac. at 5s), showing that action-aware latent dynamics encourage coherent transitions rather than optimizing each level in isolation.

Table 5: Results of Scene Initialization.

| Model | In-body | | Out-of-body | |
|---|---|---|---|---|
| | FID↓ | Recall↑ | FID↓ | Recall↑ |
| SD-3.5-Large | 275.53 | 0.0020 | 118.01 | 0.1360 |
| hunyuanimage-v2.1 | 308.46 | 0.0000 | 130.57 | 0.1520 |
| FLUX.1-dev | 324.01 | 0.0000 | 124.16 | 0.1580 |
| Qwen-Image | 260.83 | 0.0020 | 123.14 | 0.1580 |
| Gemini2.5-Flash-Image | 253.99 | 0.0000 | 172.27 | 0.0700 |
| **UniSWM** | **59.58** | **0.7960** | **104.40** | **0.2520** |

Table 6: Results of Scene Evolution.

| Model | LPIPS↓ | FID↓ | KID↓ | PSNR↑ |
|---|---|---|---|---|
| FLUX.1-Kontext-dev | 0.4734 | 81.56 | 0.0172 ± 0.0030 | 13.91 ± 2.19 |
| Qwen-Image-Edit | 0.2726 | 77.79 | 0.0193 ± 0.0041 | 17.90 ± 3.97 |
| Gemini2.5-Flash-Image | 0.3044 | 95.92 | 0.0145 ± 0.0000 | 16.61 ± 2.57 |
| **UniSWM** | **0.2251** | **49.62** | **0.0033 ± 0.0019** | **18.51 ± 3.66** |

Table 7: Fine-grained generation controlled by action and movement.

| Model | Type | Movement | | | | | Type | Action | | | | |
|---|---|---|---|---|---|---|---|---|---|---|---|---|
| | | LPIPS↓ | FID↓ | Recall↑ | PSNR↑ | SSIM↑ | | LPIPS↓ | FID↓ | Recall↑ | PSNR↑ | SSIM↑ |
| FLUX.1-Kontext-dev | Move Up | 0.4768 | 117.70 | 0.2192 | 13.84 | 0.3010 | Suture | 0.4687 | 192.81 | 0.3043 | 14.17 | 0.2978 |
| Qwen-Image-Edit | | 0.2858 | 116.28 | 0.4521 | 17.79 | 0.5632 | | 0.2897 | 197.96 | 0.4348 | 17.11 | 0.5506 |
| **UniSWM** | | **0.2305** | **78.11** | **0.5982** | **18.43** | **0.5960** | | **0.2029** | **132.67** | **0.7391** | **17.75** | **0.6590** |
| FLUX.1-Kontext-dev | Move Down | 0.4881 | 159.74 | 0.2581 | 13.71 | 0.2977 | Knotting | 0.4757 | 237.84 | 0.7692 | 12.91 | 0.2518 |
| Qwen-Image-Edit | | 0.2539 | 135.19 | 0.6022 | 18.12 | 0.5746 | | 0.2694 | 236.68 | 0.6154 | 17.72 | 0.5653 |
| **UniSWM** | | **0.2277** | **97.66** | **0.6129** | **18.46** | **0.5813** | | **0.1924** | **172.07** | **1.0000** | **19.24** | **0.6158** |
| FLUX.1-Kontext-dev | Move Left | 0.4594 | 151.51 | 0.3553 | 14.19 | 0.3031 | Dissection | 0.4845 | 109.43 | 0.1882 | 13.93 | 0.3107 |
| Qwen-Image-Edit | | 0.2910 | 157.97 | 0.5395 | 17.70 | 0.5416 | | 0.2945 | 100.47 | 0.3727 | 17.52 | 0.5366 |
| **UniSWM** | | **0.2173** | **110.75** | **0.7500** | **18.83** | **0.5696** | | **0.2504** | **70.04** | **0.4834** | **18.24** | **0.5441** |
| FLUX.1-Kontext-dev | Move Right | 0.4524 | 157.15 | 0.4426 | 14.43 | 0.3125 | Needle Puncture | 0.4606 | 157.41 | 0.1500 | 13.56 | 0.2696 |
| Qwen-Image-Edit | | 0.2681 | 163.51 | 0.5246 | 17.61 | 0.5620 | | 0.1709 | 132.58 | 0.5833 | 20.01 | 0.7195 |
| **UniSWM** | | **0.2135** | **119.71** | **0.7213** | **18.59** | **0.5885** | | **0.1345** | **79.59** | **0.6167** | **19.75** | **0.7324** |
| FLUX.1-Kontext-dev | Upper Left | 0.4790 | 175.51 | 0.3333 | 14.03 | 0.3069 | Tissue Retraction | 0.4634 | 174.05 | 0.4524 | 14.20 | 0.2963 |
| Qwen-Image-Edit | | 0.3153 | 174.47 | 0.4444 | 17.42 | 0.5350 | | 0.2880 | 171.85 | 0.5000 | 17.78 | 0.5403 |
| **UniSWM** | | **0.2521** | **126.93** | **0.7143** | **18.25** | **0.5704** | | **0.2436** | **126.12** | **0.6667** | **18.33** | **0.5535** |
| FLUX.1-Kontext-dev | Upper Right | 0.4619 | 169.14 | 0.4211 | 14.01 | 0.2980 | Needle Grasping | 0.4460 | 161.32 | 0.3824 | 13.99 | 0.2914 |
| Qwen-Image-Edit | | 0.2798 | 175.48 | 0.5263 | 18.11 | 0.5754 | | 0.2490 | 173.57 | 0.5294 | 17.30 | 0.6019 |
| **UniSWM** | | **0.2339** | **120.20** | **0.7544** | **18.77** | **0.5997** | | **0.1606** | **94.83** | **0.6471** | **18.75** | **0.7064** |
| FLUX.1-Kontext-dev | Lower Left | 0.4803 | 164.53 | 0.3030 | 13.89 | 0.3062 | Coagulation | 0.4977 | 252.17 | 0.2500 | 14.40 | 0.3296 |
| Qwen-Image-Edit | | 0.2728 | 159.37 | 0.5909 | 18.14 | 0.5592 | | 0.2796 | 219.65 | 0.5000 | 18.60 | 0.5473 |
| **UniSWM** | | **0.2278** | **110.65** | **0.7273** | **18.47** | **0.5819** | | **0.2216** | **182.22** | **0.7500** | **19.63** | **0.5586** |
| FLUX.1-Kontext-dev | Lower Right | 0.4503 | 155.49 | 0.4462 | 14.39 | 0.3103 | Aspiration | 0.4437 | 178.64 | 0.4048 | 14.05 | 0.3104 |
| Qwen-Image-Edit | | 0.2676 | 148.87 | 0.6000 | 18.21 | 0.5664 | | 0.2586 | 178.64 | 0.5476 | 18.55 | 0.5487 |
| **UniSWM** | | **0.2343** | **114.89** | **0.7538** | **18.65** | **0.5706** | | **0.2380** | **126.84** | **0.6905** | **18.69** | **0.5481** |
| FLUX.1-Kontext-dev | Move In | 0.4882 | 159.71 | 0.2600 | 13.98 | 0.3062 | Packing | 0.5279 | 427.62 | 0.6667 | 11.66 | 0.2850 |
| Qwen-Image-Edit | | 0.3009 | 160.86 | 0.4700 | 17.31 | 0.5460 | | 0.4271 | 392.75 | 0.6667 | 13.13 | 0.4327 |
| **UniSWM** | | **0.2582** | **116.78** | **0.6600** | **17.97** | **0.5613** | | **0.3729** | **394.83** | **0.6667** | **14.18** | **0.4662** |
| FLUX.1-Kontext-dev | Move Out | 0.4764 | 145.75 | 0.2925 | 13.83 | 0.3111 | **Overall** | 0.4734 | 81.56 | 0.1580 | 13.91 | 0.3014 |
| Qwen-Image-Edit | | 0.3266 | 152.49 | 0.4151 | 16.66 | 0.5066 | | 0.2726 | 77.79 | 0.3460 | 17.90 | 0.5653 |
| **UniSWM** | | **0.2763** | **113.52** | **0.5094** | **17.61** | **0.5467** | | **0.2251** | **49.62** | **0.3960** | **18.51** | **0.5859** |

## 4.3 EVALUATION OF FINE-GRAINED GENERATION

**Scene initialization.** UniSWM achieves a substantial lead for both in-body and out-of-body synthesis. For in-body initialization, FID drops to 59.58 while Recall rises to 0.7960, compared to prior systems whose FID ranges from 253.99–324.01 and Recall is at or near zero (0.0000–0.0020). Out-of-body performance also improves: FID = 104.40 and Recall = 0.2520, outperforming strong baselines such as Qwen-Image (FID = 123.14, Recall = 0.1580) and FLUX.1-dev (FID = 124.16, Recall = 0.1580). These results indicate that UniSWM initializes anatomically plausible content with higher instance coverage, rather than relying on over-smoothed or generic surgical layouts.

**Scene evolution.** On in-body editing, UniSWM achieves the lowest LPIPS (0.2251), the best FID (49.62), the lowest KID (0.0033 ± 0.0019), and the highest PSNR (18.51 ± 3.66). Compared to Qwen-Image-Edit and FLUX.1-Kontext-dev, UniSWM improves fidelity and perceptual similarity simultaneously, suggesting that the learned latent dynamics allow edits that respect local tissue appearance and lighting while following requested controls.

**Movement-conditioned control.** UniSWM consistently secures the best tradeoff across LPIPS/FID/Recall/PSNR for directed camera/tool motions (e.g., *Move Up/Down/Left/Right*, diag-

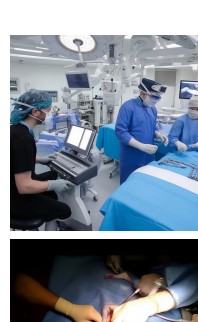 A surgeon operates a robotic surgical system from a seated position at a console. A scrub nurse stands beside the patient's bed, preparing instruments. The operating room is brightly lit with medical equipment and blue surgical drapes.

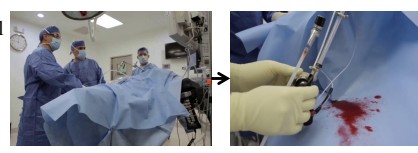 A close-up view shows gloved hands manipulating medical instruments. The instruments include a scope and tubes connected to the patient. Blood is visible on the surgical field.

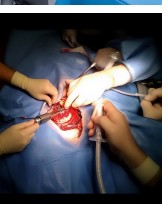 A surgical procedure is underway with a focus on the operating area. A hand holds a tool near the incision site while another hand manipulates an instrument connected to a tube. The surgical field is illuminated by a bright light.

Procedure: Dissection. Enviroment. Two tubular structures linked to the gallbladder are not clearly identified. Hepatocystic triangle is noxt cleared of fat or connective tissue, dissection does not adequately reveal the lower third of the cystic plate.

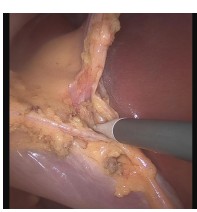

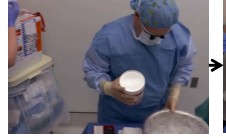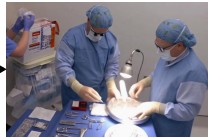 Two medical professionals are actively engaged in a surgical procedure. One is using a light source to illuminate the area of focus, while the other assists with tools. Various surgical instruments and equipment are laid out on the table beside them.

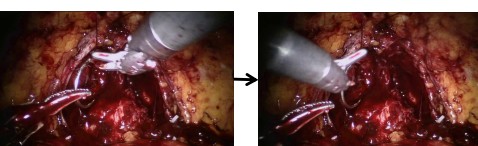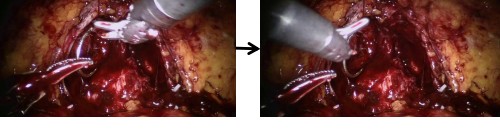

Action: Needle Grasping action; Movement: Move Left.

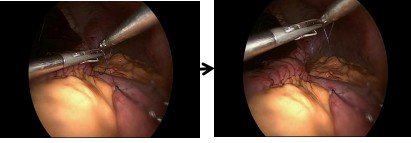

Action: Knotting; Movement: Move Up.

Figure 2: Visualization of UniSWM

onals, and *Move In/Out*). Illustratively, for *Move Up* it reaches LPIPS = 0.2305, FID = 78.11, Recall = 0.5982, PSNR = 18.43, surpassing FLUX.1-Kontext-dev and Qwen-Image-Edit on all metrics. Similar margins are observed for *Move Right* (LPIPS = 0.2135, FID = 119.71, Recall = 0.7213, PSNR = 18.59) and *Lower Right* (LPIPS = 0.2343, FID = 114.89, Recall = 0.7538, PSNR = 18.65). These gains indicate that UniSWM's latent dynamics not only predict plausible motion fields but also synthesize the corresponding texture changes without drifting from the surgical context.

**Action-conditioned control.** For fine-grained manipulations (e.g., *Suture*, *Knotting*, *Dissection*, *Needle Puncture*, *Tissue Retraction*, *Needle Grasping*, *Coagulation*, *Aspiration*), UniSWM attains the best or tied-best results across metrics. For *Knotting* it attains perfect Recall = 1.0000 with LPIPS = 0.1924, FID = 172.07, PSNR = 19.24. The improvements persist on visually confusable categories such as *Dissection* (LPIPS = 0.2504, FID = 70.04, Recall = 0.4834, PSNR = 18.24) and *Needle Puncture* (LPIPS = 0.1345, FID = 79.59, Recall = 0.6167, PSNR = 19.75). These results suggest that UniSWM's action-aware latent transitions provide reliable control signals that map textual intent to surgical kinematics and contact states.

## 5 CONCLUSION

We propose UniSWM, a unified surgical world model that couples multimodal perception with hierarchy-aligned action and movement tokens and a Mixture-of-Transformers backbone to support three capabilities within a single architecture: structured understanding, long-horizon prediction, and fine-grained, controllable visual generation across both in-body and out-of-body settings. Alongside the model, we propose UniSWM-DB and UniSWM-Bench to stress-test recognition, forecasting, and movement/action-conditioned synthesis. Empirically, UniSWM sets advanced results on present-scene understanding, leads Phase and Step prediction at all horizons, and substantially improves both scene initialization and scene evolution while adhering to fine-grained controls.

## ETHICS STATEMENT

This work involves the use of surgical video data and multimodal annotations collected from multiple institutions. All data used in this study have been de-identified to remove any personally identifiable information. The dataset, UniSWM-DB, adheres to strict privacy and ethical standards, including compliance with HIPAA and GDPR regulations, and has been anonymized to prevent re-identification of patients or personnel.

We also recognize the importance of fairness and bias mitigation. While our dataset is diverse across institutions and surgical procedures, we encourage further efforts to ensure equitable representation across demographics and surgical contexts. No harmful applications or discriminatory behaviors are intended or supported by this work.

## REPRODICIBILITY STATEMENT

To support reproducibility, we provide the following resources and methodological details:

- The UniSWM-DB dataset is described in Section 4, including annotation schemas, preprocessing steps, and data splits for temporal, cross-site, and cross-view evaluation. A data card summarizes ethical compliance, licensing, and usage guidelines is also included.

- The full architectural details of UniSWM, including the Mixture-of-Transformers design, latent dynamics modeling, and decoder specifications, are given in Section 3.

- UniSWM-Bench is introduced with task definitions, evaluation protocols, and metrics for understanding, prediction, and generation tasks. All benchmark splits and evaluation scripts are documented.

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

# A    RELATED WORK (EXTENDED)

## A.1    WORLD MODELS AND LATENT DYNAMICS

Early world models demonstrated that compact latent rollouts enable planning directly from pixels (Ha & Schmidhuber, 2018). Dreamer-style algorithms introduced stochastic latent imagination with actor–critic learning to improve stability and sample efficiency for continuous control (Hafner et al., 2019). Recent works explore scaling latent sequence modeling and credit assignment for longer horizons and complex tasks (Hao et al., 2023). In parallel, diffusion modeling—from denoising diffusion to latent diffusion—has delivered state-of-the-art image and video synthesis fidelity, offering stronger priors for forward prediction, temporal consistency, and controllability (Ho et al., 2020; Rombach et al., 2022; Cao et al., 2024). UniSWM leverages these trajectories by coupling latent dynamics for prediction with high-fidelity generative pathways for controllable synthesis.

## A.2    MULTIMODAL FOUNDATION MODELS AND VLA POLICIES

The emergence of vision–language(-action) models has connected internet-scale knowledge with embodied decision making (Ma et al., 2024). Systems such as RT-2 utilize vision–language pretraining to ground robotic policies in open-world concepts (Brohan et al., 2023), while foundation models like PaLM-E unify perception and language for downstream control (Driess et al., 2023). OpenVLA-style frameworks further emphasize generalization via modular perception–policy stacks (Kim et al., 2025). These trends suggest a path to unify recognition, prediction, and controllable synthesis, where a single backbone can interpret scenes, anticipate future states, and generate task-aligned visuals. UniSWM follows this direction but is specialized for surgical constraints, emphasizing safety, cross-view robustness, and action-/movement-aware control.

## A.3    SURGICAL INTELLIGENCE: PERCEPTION, WORKFLOW, AND CONTEXT

Surgical scene understanding covers phase/step recognition, instrument detection and segmentation, and workflow modeling (Demir et al., 2023; Ding et al., 2024a). Foundational works on endoscopic phase recognition (e.g., Cholec80) established supervised temporal modeling for intraoperative guidance (Twinanda et al., 2017; CAMMA Lab), followed by architectures that leverage spatiotemporal cues and attention (Jin et al., 2021). Community benchmarks broadened evaluation beyond in-body views: EndoVis instrument segmentation and MVOR OR datasets examine tool presence, segmentation quality, and staff/activity context under multi-view settings (Escamirosa et al., 2015; Allan et al., 2019; Srivastav et al., 2018). Despite progress, unified modeling that (i) spans in-body and out-of-body signals, (ii) performs long-horizon prediction of workflow states, and (iii) supports controllable visual synthesis remains limited.

## A.4    SURGICAL VIDEO SYNTHESIS AND CONTROLLABLE EDITING

Recent diffusion-based efforts explore surgical video generation and interactive editing (Cho et al., 2024; Iliash et al., 2024). Controllable surgical synthesis is particularly challenging: models must adhere to instrument kinematics, preserve tissue realism, and respect safety constraints while following fine-grained action or movement commands. Emerging work on multi-scale temporal prediction and incremental evolution seeks to stabilize long-horizon edits by aligning local texture updates with global intent (Zeng et al., 2025a). However, most systems still treat perception, prediction, and generation as separate modules, which can lead to temporal drift, inconsistent tool–tissue interactions, or suboptimal adherence to requested controls.

## A.5    POSITIONING OF UNISWM

UniSWM differs from prior art in three ways. *First*, it unifies structured understanding (phase, action, relations, grounding), long-horizon prediction (phase, step, and joint targets), and controllable generation (movement- and action-conditioned) within one framework, reducing cross-module mismatch. *Second*, it operates across in-body and out-of-body settings, enabling cross-view consistency and broader clinical coverage. *Third*, its generation pathway is designed to respect surgical plausibility (instrument kinematics, tissue appearance) while following fine-grained controls, addressing

common failure modes observed in prior diffusion-based pipelines (Cho et al., 2024; Iliash et al., 2024). Together with lessons from world models and VLA systems (Ha & Schmidhuber, 2018; Hafner et al., 2019; Ma et al., 2024; Brohan et al., 2023; Driess et al., 2023; Kim et al., 2025; Ho et al., 2020; Rombach et al., 2022), UniSWM targets a practical middle ground between predictive modeling and controllable synthesis tailored to surgical domains.

**Scope and Limitations.** While UniSWM advances unified modeling for surgical AI, the broader literature continues to expand in data scale, multimodal sensing (e.g., depth, audio, kinematics), and evaluation protocols. Our focus on phase/step dynamics and tool–tissue interactions complements, rather than replaces, specialized lines in registration, reconstruction, and simulation. We view UniSWM as a step toward integrated systems that couple reliable understanding and prediction with safe, user-controllable generation in real surgical workflows.

# B EXPERIMENTS

Table 8: Stability analysis of phase recognition (Max/Min/Mean across temperatures 0.5, 0.6, 0.7). Values are shown as percentages.

| Model | Accuracy Mean (Min, Max) | Recall Mean (Min, Max) | Precision Mean (Min, Max) | Jaccard Mean (Min, Max) |
|---|---|---|---|---|
| SmolVLM2-2.2B | 20.81 (20.13, 21.80) | 15.37 (13.95, 16.90) | 12.18 (11.08, 13.84) | 6.35 (5.90, 7.19) |
| Skywork-R1V-38B | 6.94 (6.81, 7.11) | 14.31 (14.24, 14.41) | 17.24 (11.27, 28.55) | 1.56 (1.31, 1.88) |
| Phi4-Multimodal | 17.92 (13.95, 20.82) | 15.50 (13.47, 17.40) | 14.14 (10.57, 19.25) | 5.36 (3.89, 6.70) |
| Mistral-Small-24B | 25.60 (23.45, 28.25) | 25.92 (23.88, 27.42) | 26.80 (23.78, 30.69) | 11.64 (10.56, 12.73) |
| PaliGemma2-3B | 7.29 (6.50, 7.75) | 13.90 (12.38, 15.42) | 10.08 (6.73, 12.10) | 2.35 (2.02, 2.75) |
| Llama-4-Scout-17B-16E | 40.70 (40.40, 41.30) | 18.02 (17.59, 18.44) | 27.96 (27.22, 29.14) | 8.71 (8.53, 8.94) |
| Kimi-VL-A3B-Instruct | 35.00 (33.60, 36.30) | 19.04 (18.27, 20.04) | 20.79 (18.35, 22.56) | 10.25 (9.57, 10.77) |
| Kimi-VL-A3B-Thinking | 6.53 (6.40, 6.60) | 14.31 (14.15, 14.43) | 16.61 (7.00, 21.52) | 1.06 (0.97, 1.16) |
| Gemma3-27B | 14.03 (14.00, 14.10) | 16.93 (16.52, 17.25) | 25.37 (15.64, 30.72) | 4.52 (4.23, 4.81) |
| MiMo-VL-7B-SFT | 20.90 (20.30, 22.10) | 16.46 (14.81, 17.91) | 20.73 (15.93, 29.04) | 7.15 (6.85, 7.40) |
| MiMo-VL-7B-RL | 25.73 (23.40, 27.90) | 14.77 (13.82, 15.44) | 15.73 (15.31, 16.45) | 6.77 (6.50, 6.98) |
| MiniCPM-V-2_6 | 16.85 (15.29, 18.56) | 14.20 (11.87, 17.09) | 14.24 (12.64, 16.09) | 6.10 (5.30, 7.30) |
| MiniCPM-o-2_6 | 15.10 (13.41, 16.40) | 20.63 (20.21, 21.08) | 23.77 (22.17, 24.70) | 8.68 (8.27, 8.94) |
| Qwen-Omni-3B | 34.16 (32.50, 36.20) | 18.27 (17.64, 18.87) | 18.11 (17.02, 19.73) | 10.59 (10.25, 10.78) |
| Qwen-Omni-7B | 22.01 (21.42, 22.50) | 18.10 (15.57, 19.39) | 18.16 (14.53, 23.07) | 7.66 (6.87, 8.26) |
| Qwen2.5-VL-7B | 27.57 (27.40, 27.90) | 18.48 (17.73, 19.46) | 20.50 (19.29, 22.06) | 8.33 (8.10, 8.60) |
| Qwen2.5-VL-32B | 44.77 (44.70, 44.90) | 22.05 (21.89, 22.18) | 26.83 (23.95, 29.77) | 13.63 (13.46, 13.83) |
| Qwen2.5-VL-72B | 37.03 (35.70, 37.80) | 20.21 (19.83, 20.85) | 23.75 (21.34, 26.03) | 10.82 (9.98, 11.48) |
| InternVL3-8B | 30.93 (30.00, 32.10) | 19.38 (18.40, 20.63) | 22.21 (17.02, 25.19) | 10.71 (9.42, 11.82) |
| InternVL3-78B | 33.17 (31.70, 34.50) | 25.25 (23.30, 26.74) | 34.01 (29.06, 42.27) | 15.05 (13.88, 15.88) |
| MedVLM-R1 | 10.50 (10.40, 10.70) | 16.15 (15.97, 16.45) | 12.54 (11.90, 13.74) | 2.40 (2.28, 2.50) |
| Lingshu-7B | 39.77 (39.60, 39.90) | 23.53 (23.22, 23.88) | 13.63 (12.36, 14.67) | 9.07 (8.84, 9.32) |
| Lingshu-32B | 36.31 (34.20, 38.70) | 22.22 (20.96, 24.44) | 23.98 (19.95, 29.19) | 12.85 (11.45, 15.10) |
| MedGemma-4B | 27.07 (25.90, 27.70) | 23.33 (22.76, 24.09) | 18.53 (17.29, 19.48) | 9.63 (9.11, 9.91) |
| MedGemma-27B | 43.40 (43.00, 43.60) | 15.30 (15.12, 15.66) | 24.68 (23.88, 25.75) | 7.37 (7.14, 7.71) |
| SurgVLM-72B | 73.05 (72.83, 73.29) | 65.30 (64.98, 65.63) | 64.91 (64.60, 65.36) | 49.10 (48.73, 49.56) |
| **UniSWM** | **79.73 (76.70, 81.90)** | **67.11 (65.07, 68.66)** | **70.54 (69.76, 71.79)** | **52.88 (50.63, 54.42)** |

Table 9: Stability analysis of action recognition (Max/Min/Mean across temperatures 0.5, 0.6, 0.7). Values are shown as percentages.

| Model | Accuracy Mean (Min, Max) | Recall Mean (Min, Max) | Precision Mean (Min, Max) | Jaccard Mean (Min, Max) |
|---|---|---|---|---|
| SmolVLM2-2.2B | 14.93 (14.70, 15.30) | 12.25 (11.69, 12.99) | 13.05 (12.20, 14.09) | 5.61 (5.50, 5.78) |
| Skywork-R1V-38B | 11.83 (11.70, 12.00) | 12.62 (12.42, 13.02) | 14.42 (13.35, 15.30) | 2.76 (2.68, 2.84) |
| Phi4-Multimodal | 24.37 (23.10, 25.10) | 12.38 (11.71, 12.88) | 13.35 (9.20, 20.92) | 5.74 (5.35, 6.02) |
| Mistral-Small-24B | 12.50 (12.40, 12.60) | 12.68 (12.57, 12.74) | 5.73 (5.47, 6.06) | 1.78 (1.76, 1.82) |
| PaliGemma2-3B | 11.57 (11.00, 12.40) | 12.58 (12.00, 13.11) | 13.92 (10.33, 17.60) | 2.26 (2.15, 2.46) |
| Llama-4-Scout-17B-16E | 30.97 (30.50, 31.20) | 12.75 (12.53, 12.90) | 19.46 (18.90, 19.99) | 4.66 (4.52, 4.79) |
| Kimi-VL-A3B-Instruct | 25.23 (24.30, 25.70) | 14.17 (12.20, 15.31) | 12.10 (10.05, 13.97) | 6.91 (5.84, 7.67) |
| Kimi-VL-A3B-Thinking | 11.23 (10.70, 11.70) | 12.90 (12.14, 13.61) | 14.07 (5.93, 23.81) | 2.33 (2.11, 2.46) |
| Gemma3-27B | 31.97 (31.90, 32.00) | 13.50 (13.47, 13.52) | 20.60 (16.43, 22.69) | 5.01 (4.97, 5.03) |
| MiMo-VL-7B-SFT | 24.37 (23.60, 25.00) | 15.72 (14.68, 17.26) | 17.56 (11.09, 25.74) | 7.40 (6.49, 8.56) |
| MiMo-VL-7B-RL | 26.90 (24.60, 28.30) | 14.04 (12.87, 14.81) | 14.21 (11.23, 18.64) | 7.05 (6.39, 7.68) |
| MiniCPM-V-2_6 | 24.43 (22.00, 26.60) | 13.42 (11.41, 15.29) | 14.14 (12.35, 16.84) | 7.41 (6.44, 8.41) |
| MiniCPM-o-2_6 | 29.67 (28.90, 30.20) | 12.47 (12.10, 12.71) | 12.60 (10.08, 14.82) | 5.22 (4.96, 5.59) |
| Qwen-Omni-3B | 28.23 (27.40, 29.10) | 12.38 (12.15, 12.80) | 12.76 (12.13, 13.08) | 5.69 (5.14, 6.12) |
| Qwen-Omni-7B | 29.93 (29.60, 30.50) | 13.20 (12.93, 13.34) | 29.58 (23.54, 39.81) | 5.81 (5.60, 6.22) |
| Qwen2.5-VL-7B | 31.17 (31.10, 31.30) | 12.55 (12.50, 12.66) | 5.82 (3.89, 9.68) | 4.02 (3.89, 4.29) |
| Qwen2.5-VL-32B | 31.67 (31.50, 31.80) | 13.72 (13.43, 14.18) | 31.84 (29.80, 35.03) | 5.34 (4.92, 5.99) |
| Qwen2.5-VL-72B | 28.83 (27.30, 30.30) | 13.42 (12.72, 14.03) | 16.52 (15.93, 17.23) | 6.21 (5.85, 6.54) |
| InternVL3-8B | 29.30 (27.80, 30.60) | 13.03 (12.34, 14.16) | 14.16 (9.65, 22.85) | 6.36 (6.12, 6.81) |
| InternVL3-78B | 28.77 (28.10, 29.70) | 12.69 (12.51, 12.98) | 21.24 (18.27, 22.93) | 6.22 (5.94, 6.39) |
| MedVLM-R1 | 31.13 (31.10, 31.20) | 12.52 (12.51, 12.55) | 16.39 (16.39, 16.40) | 3.94 (3.94, 3.95) |
| Lingshu-7B | 31.50 (31.40, 31.60) | 12.92 (12.82, 13.03) | 16.41 (16.41, 16.41) | 4.33 (4.23, 4.44) |
| Lingshu-32B | 26.60 (25.20, 27.50) | 17.96 (16.45, 19.10) | 20.07 (17.17, 21.64) | 10.09 (9.34, 10.66) |
| MedGemma-4B | 22.50 (22.30, 22.90) | 12.32 (12.08, 12.48) | 4.42 (3.74, 4.84) | 3.53 (3.41, 3.61) |
| MedGemma-27B | 29.90 (29.50, 30.30) | 12.38 (12.18, 12.58) | 20.73 (17.78, 24.71) | 4.73 (4.55, 4.88) |
| SurgVLM-72B | 45.13 (44.95, 45.35) | 31.88 (31.28, 32.46) | 30.05 (29.64, 30.79) | 18.60 (18.36, 18.91) |
| **UniSWM** | **58.53 (53.20, 63.40)** | **40.08 (35.44, 42.77)** | **49.82 (42.19, 57.44)** | **28.25 (23.57, 31.34)** |

Table 10: Stability analysis of triplet recognition (Max/Min/Mean across temperatures 0.5, 0.6, 0.7). Values are shown as percentages.

| Model | Accuracy Mean (Min, Max) | $Accuracy_{ins}$ Mean (Min, Max) | $Accuracy_{ver}$ Mean (Min, Max) | $Accuracy_{tar}$ Mean (Min, Max) |
|---|---|---|---|---|
| SmolVLM2-2.2B | 0.00 (0.00, 0.00) | 7.87 (7.50, 8.48) | 3.71 (3.23, 4.44) | 0.62 (0.35, 1.15) |
| Skywork-R1V-38B | 0.12 (0.12, 0.12) | 21.06 (20.31, 22.50) | 9.33 (9.12, 9.58) | 2.25 (1.73, 2.71) |
| Phi4-Multimodal | 0.29 (0.12, 0.46) | 11.50 (9.52, 13.10) | 12.58 (12.00, 12.98) | 3.87 (3.46, 4.10) |
| Mistral-Small-24B | 0.50 (0.29, 0.69) | 9.08 (8.42, 9.46) | 8.79 (8.60, 9.12) | 5.96 (5.48, 6.46) |
| PaliGemma2-3B | 0.00 (0.00, 0.00) | 0.96 (0.23, 1.56) | 2.19 (1.67, 3.17) | 3.06 (2.54, 3.98) |
| Llama-4-Scout-17B-16E | 0.21 (0.12, 0.29) | 3.35 (2.94, 3.58) | 3.56 (3.35, 3.75) | 7.46 (6.52, 8.02) |
| Kimi-VL-A3B-Instruct | 0.02 (0.00, 0.06) | 12.04 (9.92, 14.60) | 9.83 (9.29, 10.21) | 2.14 (2.02, 2.19) |
| Kimi-VL-A3B-Thinking | 0.04 (0.00, 0.12) | 6.50 (5.94, 6.98) | 9.21 (9.00, 9.52) | 1.08 (0.75, 1.27) |
| Gemma3-27B | 0.13 (0.12, 0.17) | 10.00 (9.81, 10.33) | 6.35 (5.94, 6.69) | 1.67 (1.38, 1.85) |
| MiMo-VL-7B-SFT | 0.13 (0.12, 0.17) | 7.98 (7.16, 8.89) | 5.10 (4.73, 5.42) | 4.85 (4.33, 5.37) |
| MiMo-VL-7B-RL | 0.21 (0.12, 0.29) | 6.25 (5.77, 6.75) | 5.14 (4.67, 5.65) | 2.14 (1.85, 2.31) |
| MiniCPM-V-2_6 | 0.04 (0.00, 0.12) | 17.02 (14.95, 18.29) | 13.93 (13.16, 14.66) | 1.35 (1.27, 1.38) |
| MiniCPM-o-2_6 | 0.35 (0.12, 0.46) | 23.31 (22.74, 24.06) | 8.92 (8.37, 9.87) | 5.29 (4.90, 5.83) |
| Qwen-Omni-3B | 0.00 (0.00, 0.00) | 10.33 (9.69, 11.02) | 7.06 (6.87, 7.21) | 1.31 (1.04, 1.50) |
| Qwen-Omni-7B | 0.15 (0.12, 0.23) | 10.79 (10.39, 11.14) | 6.87 (6.17, 7.33) | 5.46 (4.85, 6.06) |
| Qwen2.5-VL-7B | 0.23 (0.12, 0.29) | 9.92 (9.12, 10.39) | 4.65 (4.21, 4.90) | 5.21 (4.56, 6.12) |
| Qwen2.5-VL-32B | 0.23 (0.17, 0.29) | 26.22 (25.79, 26.49) | 6.69 (6.58, 6.75) | 2.10 (1.73, 2.42) |
| Qwen2.5-VL-72B | 0.19 (0.17, 0.23) | 26.33 (25.97, 26.60) | 8.16 (7.91, 8.31) | 4.58 (4.39, 4.90) |
| InternVL3-8B | 1.65 (1.56, 1.73) | 51.49 (50.72, 52.39) | 8.87 (8.71, 8.94) | 24.39 (23.72, 24.99) |
| InternVL3-78B | 0.37 (0.23, 0.52) | 37.39 (34.85, 39.35) | 8.54 (7.73, 9.41) | 3.44 (2.31, 4.50) |
| MedVLM-R1 | 0.02 (0.00, 0.06) | 45.72 (45.30, 46.28) | 8.98 (8.94, 9.06) | 0.31 (0.23, 0.40) |
| Lingshu-7B | 0.08 (0.00, 0.12) | 2.15 (1.85, 2.48) | 7.98 (7.73, 8.37) | 3.65 (3.29, 4.10) |
| Lingshu-32B | 0.21 (0.12, 0.29) | 21.04 (20.83, 21.29) | 7.17 (6.29, 8.14) | 3.77 (3.58, 3.87) |
| MedGemma-4B | 0.00 (0.00, 0.00) | 2.89 (2.65, 3.29) | 8.17 (7.67, 8.54) | 1.42 (1.15, 1.67) |
| MedGemma-27B | 0.13 (0.12, 0.17) | 8.39 (7.56, 9.29) | 3.65 (3.46, 3.87) | 3.62 (3.35, 4.10) |
| SurgVLM-72B | 4.91 (4.88, 4.96) | 47.66 (47.46, 47.78) | 12.91 (12.81, 13.01) | 38.91 (38.14, 39.41) |
| **UniSWM** | **49.73 (45.15, 55.11)** | **81.97 (80.08, 83.31)** | **70.88 (66.69, 74.90)** | **54.22 (50.67, 57.74)** |

Table 11: Stability analysis of instrument grounding (Max/Min/Mean across temperatures 0.5, 0.6, 0.7). Values are shown as percentages.

| Model | mIoU Mean (Min, Max) | mAP@0.5 Mean (Min, Max) | mAP@0.75 Mean (Min, Max) | COCO AP Mean (Min, Max) |
|---|---|---|---|---|
| SmolVLM2-2.2B | 2.89 (2.72, 3.01) | 0.27 (0.20, 0.30) | 0.07 (0.00, 0.10) | 0.08 (0.07, 0.09) |
| Skywork-R1V-38B | 9.86 (9.46, 10.10) | 1.60 (1.30, 2.00) | 0.00 (0.00, 0.00) | 0.28 (0.25, 0.32) |
| Phi4-Multimodal | 1.17 (1.05, 1.36) | 0.13 (0.00, 0.40) | 0.00 (0.00, 0.00) | 0.04 (0.00, 0.11) |
| Mistral-Small-24B | 17.64 (17.08, 18.43) | 5.93 (5.40, 6.30) | 0.27 (0.10, 0.40) | 1.45 (1.25, 1.60) |
| PaliGemma2-3B | 0.01 (0.00, 0.03) | 0.00 (0.00, 0.00) | 0.00 (0.00, 0.00) | 0.00 (0.00, 0.00) |
| Llama-4-Scout-17B-16E | 36.72 (36.45, 37.27) | 35.73 (35.50, 36.20) | 7.17 (6.50, 7.80) | 12.99 (12.92, 13.08) |
| Kimi-VL-A3B-Instruct | 9.63 (8.58, 10.94) | 4.93 (3.90, 6.10) | 0.73 (0.30, 1.30) | 1.62 (1.28, 2.23) |
| Kimi-VL-A3B-Thinking | 9.36 (9.03, 9.91) | 1.90 (1.80, 2.00) | 0.13 (0.00, 0.30) | 0.48 (0.39, 0.55) |
| Gemma3-27B | 17.45 (17.06, 17.72) | 6.17 (6.00, 6.30) | 0.10 (0.10, 0.10) | 1.41 (1.34, 1.51) |
| MiMo-VL-7B-SFT | 33.45 (32.09, 34.70) | 28.57 (26.30, 30.50) | 2.80 (2.30, 3.80) | 8.55 (7.78, 9.30) |
| MiMo-VL-7B-RL | 36.86 (36.55, 37.31) | 34.77 (33.90, 35.30) | 3.77 (3.70, 3.90) | 10.78 (10.51, 11.09) |
| MiniCPM-V-2.6 | 17.87 (16.13, 19.30) | 7.33 (5.90, 8.20) | 0.57 (0.40, 0.80) | 1.87 (1.47, 2.14) |
| MiniCPM-o-2.6 | 20.33 (18.70, 22.43) | 9.80 (8.20, 12.30) | 1.10 (0.80, 1.40) | 2.96 (2.45, 3.85) |
| Qwen-Omni-3B | 22.17 (20.52, 23.61) | 16.20 (14.10, 18.00) | 3.47 (2.30, 4.70) | 6.25 (5.06, 7.44) |
| Qwen-Omni-7B | 34.60 (33.53, 35.68) | 40.53 (38.30, 42.10) | 5.63 (4.70, 6.60) | 14.17 (13.35, 15.23) |
| Qwen2.5-VL-7B | 10.63 (10.44, 10.84) | 0.97 (0.60, 1.30) | 0.03 (0.00, 0.10) | 0.26 (0.20, 0.32) |
| Qwen2.5-VL-32B | 11.36 (11.13, 11.55) | 1.93 (1.60, 2.30) | 1.07 (0.90, 1.30) | 1.03 (0.87, 1.24) |
| Qwen2.5-VL-72B | 42.70 (42.67, 42.72) | 45.95 (45.60, 46.30) | 25.75 (25.20, 26.30) | 25.69 (25.25, 26.13) |
| InternVL3-8B | 22.49 (22.14, 22.81) | 7.13 (6.70, 7.80) | 0.20 (0.10, 0.30) | 1.58 (1.45, 1.78) |
| InternVL3-78B | 29.41 (28.38, 30.39) | 18.20 (17.40, 18.80) | 1.60 (1.20, 2.00) | 4.99 (4.60, 5.20) |
| MedVLM-R1 | 5.31 (5.29, 5.33) | 0.30 (0.30, 0.30) | 0.00 (0.00, 0.00) | 0.03 (0.03, 0.03) |
| Lingshu-7B | 28.72 (28.67, 28.80) | 26.13 (25.80, 26.50) | 3.37 (2.70, 3.80) | 8.20 (7.92, 8.37) |
| Lingshu-32B | 26.03 (24.72, 27.01) | 18.20 (16.90, 19.70) | 2.90 (2.40, 3.70) | 5.89 (5.40, 6.58) |
| MedGemma-4B | 9.43 (9.20, 9.70) | 0.57 (0.40, 0.80) | 0.00 (0.00, 0.00) | 0.08 (0.02, 0.12) |
| MedGemma-27B | 17.20 (17.08, 17.36) | 2.23 (2.10, 2.40) | 0.00 (0.00, 0.00) | 0.44 (0.39, 0.48) |
| SurgVLM-72B | 59.34 (59.14, 59.50) | 74.20 (73.30, 74.80) | 28.30 (27.90, 28.90) | 36.28 (36.13, 36.46) |
| **UniSWM** | **86.41 (84.63, 88.11)** | **92.90 (92.00, 94.20)** | **87.43 (85.70, 88.70)** | **69.76 (65.97, 73.75)** |

Table 12: Stability analysis of environment answering (Max/Min/Mean across temperatures 0.5, 0.6, 0.7). Values are shown as percentages.

| Model | Accuracy Mean (Min, Max) | BLEU-4 Mean (Min, Max) | METEOR Mean (Min, Max) | ROUGE-1 Mean (Min, Max) |
|---|---|---|---|---|
| SmolVLM2-2.2B | 23.65 (22.22, 25.64) | 3.03 (1.80, 4.73) | 12.19 (5.26, 21.15) | 15.60 (7.92, 25.06) |
| Skywork-R1V-38B | 35.63 (34.90, 36.41) | 0.32 (0.26, 0.37) | 1.36 (1.23, 1.54) | 1.56 (1.35, 1.73) |
| Phi4-Multimodal | 34.25 (34.03, 34.55) | 0.34 (0.33, 0.36) | 1.48 (1.41, 1.58) | 1.74 (1.67, 1.84) |
| Mistral-Small-24B | 36.79 (36.13, 37.34) | 0.15 (0.11, 0.17) | 0.59 (0.52, 0.64) | 0.71 (0.58, 0.78) |
| PaliGemma2-3B | 31.78 (31.61, 31.99) | 1.55 (1.36, 1.71) | 5.65 (4.78, 6.18) | 9.04 (7.94, 9.78) |
| Llama-4-Scout-17B-16E | 36.99 (36.54, 37.23) | 0.04 (0.03, 0.04) | 0.20 (0.18, 0.22) | 0.33 (0.31, 0.36) |
| Kimi-VL-A3B-Instruct | 32.11 (31.81, 32.66) | 1.65 (1.50, 1.81) | 6.62 (6.09, 7.17) | 8.60 (7.95, 9.34) |
| Kimi-VL-A3B-Thinking | 34.09 (33.87, 34.24) | 0.34 (0.32, 0.36) | 1.44 (1.40, 1.49) | 1.74 (1.68, 1.78) |
| Gemma3-27B | 36.02 (35.76, 36.34) | 0.03 (0.03, 0.03) | 0.13 (0.13, 0.13) | 0.26 (0.24, 0.29) |
| MiMo-VL-7B-SFT | 35.40 (34.77, 36.34) | 0.09 (0.06, 0.11) | 0.37 (0.26, 0.48) | 0.66 (0.53, 0.73) |
| MiMo-VL-7B-RL | 34.97 (34.48, 35.70) | 0.10 (0.09, 0.11) | 0.40 (0.37, 0.43) | 0.74 (0.66, 0.85) |
| MiniCPM-V-2.6 | 33.32 (31.96, 35.45) | 1.14 (1.01, 1.21) | 5.50 (4.28, 6.52) | 6.18 (5.36, 6.81) |
| MiniCPM-o-2.6 | 35.29 (34.85, 35.91) | 0.24 (0.22, 0.27) | 0.74 (0.71, 0.76) | 1.15 (1.06, 1.21) |
| Qwen-Omni-3B | 35.03 (34.01, 35.58) | 0.14 (0.12, 0.15) | 0.87 (0.81, 0.97) | 0.73 (0.63, 0.82) |
| Qwen-Omni-7B | 37.45 (37.21, 37.80) | 0.26 (0.24, 0.29) | 1.06 (0.95, 1.25) | 1.24 (1.15, 1.37) |
| Qwen2.5-VL-7B | 37.17 (35.87, 38.82) | 4.43 (3.73, 4.83) | 15.14 (13.10, 16.53) | 13.27 (12.00, 14.01) |
| Qwen2.5-VL-32B | 42.43 (42.04, 42.81) | 0.21 (0.20, 0.22) | 2.08 (2.00, 2.13) | 1.74 (1.70, 1.78) |
| Qwen2.5-VL-72B | 41.89 (41.82, 41.97) | 0.24 (0.20, 0.27) | 1.62 (1.50, 1.69) | 2.37 (2.11, 2.57) |
| InternVL3-8B | 34.77 (34.33, 35.54) | 0.98 (0.89, 1.08) | 4.37 (4.08, 4.59) | 4.35 (4.05, 4.62) |
| InternVL3-78B | 36.70 (36.54, 36.85) | 0.16 (0.13, 0.17) | 1.39 (1.39, 1.40) | 0.92 (0.88, 0.97) |
| MedVLM-R1 | 31.25 (30.68, 31.73) | 2.42 (2.39, 2.45) | 7.66 (7.61, 7.74) | 10.85 (10.75, 11.04) |
| Lingshu-7B | 35.41 (34.71, 36.04) | 0.71 (0.69, 0.75) | 2.74 (2.65, 2.85) | 3.02 (2.93, 3.18) |
| Lingshu-32B | 34.69 (34.28, 35.14) | 0.11 (0.10, 0.12) | 0.59 (0.58, 0.60) | 0.74 (0.69, 0.83) |
| MedGemma-4B | 37.09 (36.36, 37.48) | 0.05 (0.03, 0.06) | 0.32 (0.27, 0.38) | 0.58 (0.52, 0.64) |
| MedGemma-27B | 35.98 (34.86, 36.63) | 0.05 (0.04, 0.06) | 0.24 (0.18, 0.30) | 0.44 (0.37, 0.50) |
| SurgVLM-72B | 69.52 (69.21, 69.75) | 46.65 (46.34, 46.91) | 58.16 (57.83, 58.45) | 74.58 (74.16, 74.92) |
| **UniSWM** | **65.98 (62.85, 69.85)** | **92.14 (91.55, 92.83)** | **96.98 (96.77, 97.24)** | **93.56 (93.16, 94.16)** |

## C    LARGE LANGUAGE MODELS USAGE STATEMENT

In this study, LLMs were not involved in any content creation. Their use was limited strictly to language polishing. The complete manuscript draft was produced by the authors, with LLMs only used to optimize English grammar and improve the clarity of expression.

