# OpenReview forum: "Unified Surgical World Model for Structured Understanding, Long-Horizon Prediction, and Fine-Grained Generation"
_ICLR.cc/2026/Conference — ICLR 2026 Conference Desk Rejected Submission_

### Official Review · Reviewer_U8Tf · 2025-10-26

**Soundness:** 2
**Presentation:** 2
**Contribution:** 2
**Rating:** 4
**Confidence:** 4

**Summary:**

This paper introduces a Unified Surgical World Model (UniSWM) capable of understanding multiple surgical tasks, predicting surgical phases and steps over a time horizon, and generating surgical frames conditioned on actions. A surgical multi-modal dataset is curated for this study, consisting primarily of intra-operative (inside-the-body camera views) samples, along with out-of-the-body (operating room) views. This dataset serves as a benchmark to evaluate the proposed model against baselines on different surgical tasks.

**Strengths:**

1. The proposed framework, UniSWM, presents a new and innovative approach within the surgical context. The unification of surgical context understanding, prediction of surgical actions over a given time horizon, and generation of surgical frames conditioned on actions into a single model could be beneficial for the surgical machine learning audience.

2. The curated dataset, UniSWM-DB, comprising 1.81 million samples, is a valuable contribution to the data-scarce surgical community. Although some clarification about the dataset is needed, the paper’s contribution would be even more significant if the dataset and its details were made publicly available to the research community.

3. The evaluation is performed on multiple relevant and challenging tasks in the surgical domain, including surgical phase, step, and action recognition, tool detection, caption generation, and image generation. There exists a lack of clinically relevant text captioning models and this work could be useful to enhance and generate fine granular text of surgical scenes (intricate information regarding a scene).

**Weaknesses:**

The weaknesses are briefly outlined below. Each point has corresponding questions raised in the Questions section.


1. Motivation of the work: The motivation for developing a Surgical World Model is not sufficiently justified. While the paper states that such models can facilitate multi-modal data generation for robotic simulators or training VLMs, these applications are not explored or validated in the experiments. A more concrete and well-aligned motivation is necessary to establish the relevance and impact of the proposed framework.


2. Clarity in the methods section: The methods section introduces several new variables and notations without adequate explanation. Better clarity is required to clearly define the model inputs, outputs, and training objectives.

3. Information about the dataset: The description of the curated UniSWM-DB dataset lacks sufficient detail. Important aspects such as dataset composition, data sources, annotation process, and train/test splits are not clearly presented. This lack of transparency raises concerns regarding the validity of the reported results.

4. Evaluation: The choice of baselines used for comparison is not well justified, and the evaluation methodology lacks sufficient detail. Specifically, the criteria and experimental setup for prediction and generation tasks are not clearly described, making it difficult to assess the performance and generalizability of the proposed model.

**Questions:**

Q1. Motivation of the work


1.1 The paper mentions that data generated from Surgical World Models can serve as simulators or be used for training other downstream models (lines 46–49, 61–63). For instance, the generated data could be used to train a VLM or as simulated images to train surgical policy models, thereby demonstrating their utility in downstream tasks. However, if such evaluations are not performed, the paper should refrain from making these claims and instead narrow its scope to “evaluating the capability of the UniSWM model for surgical tasks such as …”.


Q2. Methods section


2.1 The “initial scene generation” step in Section 3 is not clearly explained. How does this differ from the text encoder and the tokenization process from the text prompt shown in Fig. 1? The significance of this step is crucial to understand the different components of the proposed UniSWM model.


2.2 Line 145 mentions that visual tokens are provided as input to an LLM, but no LLM is introduced in this section. What specific LLM is used, and how are the visual tokens processed before entering the mixture-of-experts transformer blocks?


2.3 Line 185 mentions teacher forcing. If this concept is adopted from prior literature, an appropriate citation should be provided. If not, the term needs to be explained in more detail to aid understanding of the scene analysis section.


2.4 The training objective described in line 219 requires elaboration. The individual loss terms should be mathematically defined, at least in the appendix, to enable readers to fully understand the training process.


Q3. Clarity on the benchmark dataset

3.1 What surgical procedures are included in the curated dataset? What are the specific phases, steps, and actions represented? The information provided is insufficient. Additionally, what does “out-of-the-body caption control” refer to, and what are the “out-of-the-body” tasks included in the dataset?

3.2 The dataset reportedly contains 1.81 million samples. What constitutes a sample — an image frame, a video, or a patient? How many patients and procedures are included? How diverse is the dataset in terms of hospitals, surgeon expertise, or patient demographics? These details are essential to assess the generalizability of the model.

3.3 What data splitting strategy was used (train/validation/test)? Given that the number of samples per task ranges between 500 and 4000, how many were allocated to the test set? Does the dataset exhibit class imbalance? If so, how was this issue addressed?


3.4 The paper states that only the generation task includes out-of-body surgical contexts. This appears to exaggerate the contribution, as it implies the dataset comprehensively covers both intra-operative and out-of-body data.

Q4. Evaluation

4.1 What is the motivation for including 29 different models when only two strong VLMs are relevant (line 260)? The evaluation in Tables 3 and 4 appears misleading, as zero-shot models are compared against UniSWM, which is trained on the benchmark dataset. This comparison is unfair since UniSWM already has been trained on the dataset, whereas baseline VLMs have not been trained in the surgical domain. For a fair comparison, fine-tuning a subset of these models on the same dataset would provide a more meaningful assessment. Without such experiments, the work mainly presents a zero-shot evaluation rather than a strong demonstration of the proposed architecture’s advantages.


4.2 What is the purpose of the image generation task? The generated images are evaluated only for visual quality against baselines not trained on surgical datasets. Given the known sensitivity of diffusion models to different lighting conditions [1] (common in surgical datasets), fine-tuning these models on surgical data would be necessary for a fair comparison.

4.3 Only two images are generated for the prediction task, even though the model is conditioned on multiple frames of surgical phases and steps. Predicting only frames t+1 and t+H (H being the time at a certain point) does not provide insight into temporal consistency. What happens to the intermediate frames? Why was such a strategy used for long horizon prediction?

4.4 Without further dataset details, the model seems likely to overfit the proposed benchmark. Depending on the type of surgical procedures, the model should be evaluated—at least in a zero-shot fashion—on open-source datasets such as Cholec80, SAR-RARP50, or GraSP, GynSurg [2] to better demonstrate generalizability.

4.5 Ablation study of the different loss terms is lacking. Does the unification of understanding, prediction and generation have inter-dependencies ? These insights can be revealed with an ablation study accompanied with the suggested downstream tasks (see below).


Minor comments

a. The first few paragraphs of the introduction largely repeat the abstract. While this is acceptable, restructuring them to emphasize why multi-modal datasets are important for surgical research—and the challenges in obtaining surgical data—would strengthen the motivation. The introduction could also better highlight the potential applications of surgical VLMs.

b. The related work section mentions that prior works fail to maintain instrument kinematics and tissue realism. How does the proposed approach address or quantify these aspects? If not directly addressed, this claim should be revised, and relevant citations added.

c. Numerous baseline models are used in this study. Please list and cite all of them in the appendix to acknowledge prior work.

d. Table 1 is not referenced in the main text.

e. There are a lot of new work on surgical video generation beyond that mentioned in the related work section in the appendix. The paper can include these new works (SurGen, S2GVID, HierSurg, Ophora) and compare how their generation component differs so that it strengthen their work.

Suggestions (not necessary for the rebuttal and final paper decision)

1. The generated images could also be evaluated in downstream tasks such as classification, segmentation, or tool detection. For example, given an image with a surgical tool and an action, one could evaluate tool position via segmentation or detection, thereby demonstrating the practical value of image generation. Merely assessing visual quality does not sufficiently justify the inclusion of the generation module. See eval. Section of works such as [3,4,5].


2. Beyond static images, video generation could also be explored using a video decoder. The generated videos could then be evaluated on downstream tasks like phase prediction or tool trajectory estimation to assess temporal coherence. Actions such as suturing, knotting, or dissection involve multiple frames, and evaluating only single-frame quality is insufficient. Refer to [6,7,8] for how surgical videos can be used for different surgical video vision tasks.

References


1. Lin et al, Common Diffusion Noise Schedules and Sample Steps are Flawed, 2024.

2. Cholec80: Twinanda et al, EndoNet: A Deep Architecture for Recognition Tasks on Laparoscopic Videos, 2017; SAR-RARP50: Psychogyios et al., Segmentation of surgical instrumentation and Action Recognition on Robot-Assisted Radical Prostatectomy Challenge, 2024; GraSP: Ayobi et al, Pixel-wise Recognition for Holistic Surgical Scene Understanding,2025; GynSurg: Sahar et al., A Comprehensive Gynecology Laparoscopic Surgery Dataset, 2025.

3. Nwoye et al, Surgical text-to-image generation, 2024

4. Cho et al., Surgen: Text-guided diffusion model for surgical video generation, 2024

5. Paul et al., Sim-To-Real Transfer for Visual Reinforcement Learning of Deformable Object Manipulation for Robot-Assisted Surgery, 2024

6. Li et al., Endora: Video Generation Models as Endoscopy Simulators, 2024

7. Venkatesh et al., Mission Balance: Generating Under-Represented Class Samples Using Video Diffusion Models, 2025

8. Sivakumar et al., SG2VID: Scene Graphs Enable Fine-Grained Control for Video Synthesis, 2025

---

### Official Review · Reviewer_MzVj · 2025-10-27

**Soundness:** 2
**Presentation:** 2
**Contribution:** 3
**Rating:** 4
**Confidence:** 5

**Summary:**

This work explores surgical world model in a unified way. They unify the high-level perceptual tasks and low-level generation tasks in an architecture to acts as both a data generator and a simulator for surgical robotics. A dataset benchmark is proposed to better train and evaluate the surgical world model.

**Strengths:**

1.	It’s  great to introduce a comprehensive benchmark to evaluate the surgical world model, covering understanding, prediction, and generation tasks.
2.	The first unified model to integrates structured understanding, long-horizon prediction, and fine-grained generation
3.	The quantitative comparisons are sufficient, including 29 existing models.

**Weaknesses:**

1.	The literature of world models is insufficient. There have been already some world models in medical/surgical domain, such as [1][2]. The author should present their novelty compared to existing work.
2.	The details of training stage are missing. While the objectives of understanding, prediction, and generation are controlled by hyper-parameter weights, it is hard to find accurate weights to well balance the high-level and low-level tasks. Please clarify this. Also, which parts of model are frozen or finetuned?
3.	The details of the proposed dataset are missing. While the author presented the distribution of the dataset, they did not describe the annotation pipeline. The proposed method tried to forecast multi-step Phase and Step trajectories, the training must require tensive annotation. Please clarify how to collect them and cite the source datasets if used.
4.	There are many popular works unifying understanding and generation. However, an important bottleneck is how to organize low-level latents and high-level latents using VAE. While the author presented the architectures of visual encoder, they did not explain why such design is better.
5.	Why we need unified surgical world model? The ablations are missing. The author should validate if understanding really helps generation, and generation improves understanding in their method. Is a unified method superior to separate ones?
6.	Are these general VLMs finetuned on the proposed dataset? If not, the experiments may be not fair.
7.	It would be better to present the visual comparison of generation against existing methods.

[1] Yang Y, Wang Z Y, Liu Q, et al. Medical world model: Generative simulation of tumor evolution for treatment planning[J]. arXiv preprint arXiv:2506.02327, 2025.
[2] Koju S, Bastola S, Shrestha P, et al. Surgical Vision World Model[C]//MICCAI Workshop on Data Engineering in Medical Imaging. Cham: Springer Nature Switzerland, 2025: 1-10.

**Questions:**

Please see the weaknesses

---

### Official Review · Reviewer_A1Zn · 2025-10-30

**Soundness:** 3
**Presentation:** 2
**Contribution:** 2
**Rating:** 4
**Confidence:** 3

**Summary:**

The paper proposes UniSWM, a transformer-based model for unified surgical understanding, prediction, and generation. It introduces a new multimodal dataset (UniSWM-DB) and benchmark (UniSWM-Bench) and shows improvements over strong baselines across various surgical tasks.

**Strengths:**

1) The work makes a ambitious attempt to unify surgical understanding, long-horizon prediction, and controllable generation within one coherent framework—an important step toward comprehensive surgical AI.
2) The Mixture-of-Transformers backbone and hierarchical token design (phase, step, action, movement) are thoughtfully crafted to capture the structured nature of surgical workflows.
3) Experiments span multiple subtasks and datasets, showing consistent gains across understanding, prediction, and generation benchmarks. Figures and examples effectively illustrate UniSWM’s capability in fine-grained comprehension and token-controlled scene generation.
4) The release of UniSWM-DB and UniSWM-Bench could provide valuable resources for the surgical AI community.

**Weaknesses:**

1) The methodology lacks sufficient detail — the model architecture, especially the integration of transformer components, is unclear. Several methodological details (token hierarchy, routing, decoder training) are underexplained, reducing reproducibility.
2) The comparison may not be fair, as other VLMs were not trained on surgical data, whereas the proposed method is trained on your own UniSWMDB dataset.
3) It is unclear whether this truly qualifies as a “world model,” since both training and testing are conducted on the proposed UniSWMDB dataset.
4) Limited comparison with related transformer-based surgical works, making novelty claims less convincing. Furthermore, fairness of comparison unclear—gains may result from dataset scale rather than model design. Dataset diversity, ethics, and generalization aspects are insufficiently discussed.
5) Figures and tables do not clearly map pipeline components to mathematical formulations.

**Questions:**

Refer to the weakness please

---

### Official Review · Reviewer_82oS · 2025-10-30

**Soundness:** 2
**Presentation:** 1
**Contribution:** 2
**Rating:** 2
**Confidence:** 4

**Summary:**

UniSWM introduces a unified surgical world model that couples multimodal perception with a Mixture-of-Transformers to support three capabilities in one architecture: structured understanding, long-horizon prediction, and fine-grained, controllable visual generation across both in-body and OR scenes. The model encodes observations into a latent state and advances it with hierarchy-aligned discrete tokens (Phase/Step/Action/Movement), enabling future state forecasting and token-conditioned frame synthesis—without optical flow or kinematic labels. To train and evaluate at scale, the authors release UniSWM-DB ($(\approx 1.81,\text{M})$ samples) and UniSWM-Bench (5 understanding, 2 prediction, 3 generation tasks). Empirically, UniSWM outperforms strong baselines (e.g., GPT-5, Gemini-2.5-Pro, Qwen-VL-Max), leading Phase/Step forecasting at horizons $(H\in{1,5,30,60},\text{s})$ and delivering large gains in scene initialization/evolution and in action/movement-conditioned control, showing that token-driven latent dynamics yield coherent, controllable surgical video while improving recognition and forecasting.

**Strengths:**

**Original:**
This work integrates structured understanding, long-horizon prediction, and fine-grained controllable generation into a unified surgical-world model that operates across in vivo and operating-room (OR) perspectives. It organizes control via hierarchically aligned discrete tokens (Phase/Step/Action/Movement) and achieves controllable synthesis without requiring optical flow or kinematic annotations. At the same time, it provides sufficiently comprehensive experments.

**Quality:**
Methodologically, the authors adopt a Mixture-of-Transformers backbone with a clear division of labor among three decoders for understanding, prediction, and generation, providing an end-to-end pipeline from textual state modeling to latent-space planning and conditional decoding; experiments cover multi-task understanding, prediction, and generation, and demonstrate a significant lead compared to strong commercial and academic baselines.

**Significance:**
This unified framework functions both as a data generator and an embodied simulator, directly alleviating the bottlenecks of data scarcity and the lack of high-fidelity simulators in the surgical domain, and comprehensively enhancing understanding, long-term prediction, and controllable generation.

**Weaknesses:**

**Compute, latency, and deployment details are missing.**
The paper outlines losses and modules but omits training budgets, hardware, inference latency, and memory footprint—key for OR deployment. Include training hours/GPUs, and real-time feasibility analysis.

**Evaluation is centered on an in-house benchmark; generalization beyond it is unclear.**
All core results come from UniSWM-DB/Bench; there’s no external validation on widely used public datasets (e.g., Cholec80, EndoVis, MVOR) that are cited in Related Work. This raises domain-shift concerns and makes it hard to compare with prior arts’ published numbers. Add cross-dataset tests (zero-/few-shot) and report cross-site splits explicitly.

**Bias and representativeness are acknowledged but unresolved.**
The Ethics section “encourage further efforts to ensure equitable representation,” suggesting current coverage may be uneven. Provide demographic/procedure breakdowns, fairness audits across hospitals and device vendors, and mitigation strategies.

**Clinical/safety validity of generated content is weakly measured.**
Generation is judged by FID/LPIPS/PSNR/Recall, which do not capture surgical plausibility (e.g., valid tool–tissue contacts, respect of anatomical constraints). Add surgeon-in-the-loop ratings, kinematic-consistency metrics (tool trajectory smoothness, contact events), and safety-critical checklists tailored to the “Environment/Safety” heads.

**Questions:**

# Questions
1. How to process the data to construcet the UniSWM-DB and UniSWM-Bench? What is the Medical Category Composition (not Generation, Prediction or Understanding) of the Dataset？Could aurhors provide more data description and samples? How many hours or just images?

2. Does Understanding and Prediction Decoder use any LLM base model and Generation Decoder use any base model? Could authors provide more detailed visulization samples of the whole pipeline?

3. Could you add ablations for (a) **MoT vs. single-branch Transformer**, (b) **with/without discrete movement tokens**, (c) **generation head: flow-matching vs. autoregressive**, and (d) **decode-only-when-needed** vs. dense decoding? This would clarify **where the gains come from**.

4. Could authors provide more time cost and GPU cost for inference? This is important for its application.

---

### Note · Program_Chairs · 2026-01-17
**Submission Desk Rejected by Program Chairs**

The following references in this submission do not refer to real documents and/or have major errors in bibliographic information:

 Minjong Kim, Chenfeng Meng, Di Zhang, et al. OpenVLA: An open-source vision-language-action model. In Proceedings of the International Conference on Machine Learning (ICML), volume 270 of Proceedings of Machine Learning Research. PMLR, 2025.